# Contact-independent killing mediated by a T6SS effector with intrinsic cell-entry properties

Li Song [1,4], Junfeng Pan [1,4], Yantao Yang [1,4], Zhenxing Zhang [1], Rui Cui [1], Shuangkai Jia [1], Zhuo Wang[1], Changxing Yang [1], Lei Xu [1], Tao G. Dong[2,3], Yao Wang [1✉] & Xihui Shen [1✉]

Bacterial type VI secretion systems (T6SSs) inject toxic effectors into adjacent eukaryotic and prokaryotic cells. It is generally thought that this process requires physical contact between the two cells. Here, we provide evidence of contact-independent killing by a T6SS-secreted effector. We show that the pathogen *Yersinia pseudotuberculosis* uses a T6SS (T6SS-3) to secrete a nuclease effector that kills other bacteria in vitro and facilitates gut colonization in mice. The effector (Tce1) is a small protein that acts as a $Ca^{2+}$- and $Mg^{2+}$-dependent DNase, and its toxicity is inhibited by a cognate immunity protein, Tci1. As expected, T6SS-3 mediates canonical, contact-dependent killing by directly injecting Tce1 into adjacent cells. In addition, T6SS-3 also mediates killing of neighboring cells in the absence of cell-to-cell contact, by secreting Tce1 into the extracellular milieu. Efficient contact-independent entry of Tce1 into target cells requires proteins OmpF and BtuB in the outer membrane of target cells. The discovery of a contact-independent, long-range T6SS toxin delivery provides a new perspective for understanding the physiological roles of T6SS in competition. However, the mechanisms mediating contact-independent uptake of Tce1 by target cells remain unclear.

[1] State Key Laboratory of Crop Stress Biology for Arid Areas, Shaanxi Key Laboratory of Agricultural and Environmental Microbiology, College of Life Sciences, Northwest A&F University, 712100 Yangling, Shaanxi, China. [2] State Key Laboratory of Microbial Metabolism, Joint International Research Laboratory of Metabolic & Developmental Sciences, School of Life Sciences and Biotechnology, Shanghai Jiao Tong University, 200240 Shanghai, China. [3] Department of Ecosystem and Public Health, University of Calgary, Calgary, AB T2N 4Z6, Canada. [4]These authors contributed equally: Li Song, Junfeng Pan, Yantao Yang. ✉email: wangyao@nwsuaf.edu.cn; xihuishen@nwsuaf.edu.cn

To survive in complex microbial communities such as the intestinal microbiota, bacteria have evolved various molecular weapons to compete with other species. The classic examples of these weapons are diffusible antimicrobials such as broad-spectrum antibiotics and strain-specific bacteriocins (including microcins, bacteriocins of less than 10 kDa produced through the ribosomal pathway), which can exert long-range inhibitory effects on target cells[1–4]. Some gram-negative bacteria use contact-dependent growth inhibition (CDI), composed by CdiB translocators and CdiA toxins, to compete with other bacteria. Similar to bacteriocins and microcins, CDI requires specific outer membrane (OM) receptors on susceptible bacteria for translocation of CdiA toxins into the target cell[5,6]. The type VI secretion system (T6SS) is another contact-dependent weapon in the bacterial competition arsenal but its delivery is generally considered independent of receptors[7].

T6SSs are widely distributed transmembrane complexes used by many gram-negative bacteria to translocate effectors into adjacent cells in a contact-dependent manner[8–11]. Although some T6SSs are involved in bacterial pathogenesis through delivery of anti-eukaryotic effectors into host cells[12–14], T6SS is primarily considered an antibacterial weapon to compete against rival bacteria in polymicrobial environments[9,15,16]. The antibacterial function of T6SSs relies on injection of antibacterial effectors that target conserved, essential features of the bacterial cell. Each antibacterial effector is co-expressed with a cognate immunity protein, which protects the producing cells from self-intoxication[17,18]. All contact-dependent T6SS antibacterial weapons characterized to date do not require specific receptors in target cells for delivery of effectors or recognition of prey cells[4,7]. Recently, Si and colleagues[19,20] reported a contact-independent role of T6SS in metal acquisition through secretion of metal-scavenging effectors into the extracellular milieu. These findings raise the question of whether T6SS can similarly secrete toxic effectors into the extracellular milieu that recognize and enter target cells, thereby mediating contact-independent killing.

In this work, we report that the *Yersinia pseudotuberculosis* (*Yptb*) T6SS-3 mediates a contact-independent killing pathway through secretion of a nuclease effector, Tce1, into extracellular medium. Unlike canonical T6SS effectors, Tce1 can enter prey cells after being secreted into the medium. The Tce1-mediated T6SS antibacterial pathway is also required for optimal colonization of the mouse gut by *Yptb*, whereby it eliminates competing commensals and enteric pathogens. The discovery of a contact-independent T6SS killing pathway adds to current knowledge of the role of T6SS in competition, and opens a new avenue for understanding the ecological consequences of T6SS.

## Results

**Identification of novel T6SS effectors**. To identify novel T6SS effectors, we searched the *Yptb* YPIII genome for genes containing the Proline-Alanine-Alanine-aRginine (PAAR) domain, a conserved effector-targeting domain that is linked or adjacent to numerous known T6SS effectors[7,21,22]. A gene locus encoding multiple hypothetical T6SS effector-immunity pairs was identified (YPK_0952-0958, Fig. 1a). Both the first and last open-reading frame (ORF) of this locus contain PAAR domains. The first ORF, YPK_0952, contains a typical PAAR domain at its N-terminus and an S-type pyocin domain at its C-terminus. When VSVG-tagged YPK_0952 was produced in YPIII, the secreted protein was readily detected in the supernatant. However, YPK_0952 secretion was abrogated in the Δ4*clpV* mutant, in which all four essential ATPase genes in the four sets of T6SSs were deleted, strongly suggesting that YPK_0952 is a T6SS

effector. The secretion of YPK_0952 was dramatically diminished with deletion of *clpV3*, but not with deletion of *clpV1*, *clpV2*, or *clpV4*, further indicating that YPK_0952 is a T6SS effector mainly associated with T6SS-3 (Supplementary Fig. 1a). Similarly, we showed that YPK_0954, which does not contain a PAAR domain but is located downstream of YPK_0952, is also a T6SS-3 effector (Fig. 1b and Supplementary Fig. 1b).

To confirm the toxic activity of YPK_0954, we performed toxicity assays in *Escherichia coli*. Expression of YPK_0954, a 67-amino acid (aa) protein in *E. coli* results in significant growth inhibition. This growth inhibition was relieved by co-expression of the immediately downstream gene YPK_0955, which encodes a protein containing the colicin_immun domain in the *ypk_0954-ypk_0955* bicistron (Fig. 1c). This result suggests that YPK_0955 is the cognate immunity protein for YPK_0954. We renamed YPK_0954 as T6SS contact-independent antibacterial effector 1 (Tce1) and the immunity protein YPK_0955 as T6SS contact-independent antibacterial immunity 1 (Tci1) for reasons described below.

To assess whether the immunity results from direct protein-protein interaction, we performed glutathione S-transferase (GST) pull-down and bacterial two-hybrid assays, and the results showed specific interactions between Tce1 and Tci1 (Fig. 1d, e). These results indicate that Tce1 is a T6SS-3 secreted antibacterial effector and that its toxicity is neutralized by the Tci1 immunity protein.

**Tce1 exhibits Ca²⁺, Mg²⁺-dependent DNase activity**. Having demonstrated that Tce1 is a T6SS effector, we sought to investigate its biochemical activity. No predictable functional domain could be identified in Tce1 using the BLASTP search or other bioinformatics tools. However, further analysis using HHpred[23] revealed similarity of Tce1 with the DNA-binding proteins BldC in *Streptomyces coelicolor* and CedA in *E. coli*, implying its potential role as a nuclease toxin. Incubation of chromatography-purified Tce1 (Supplementary Fig. 1c) with λ-DNA in the same reaction buffer as DNase I led to dramatic DNA degradation in a pattern similar to DNase I (Fig. 1f). The DNase activity of Tce1 critically relies on the co-existence of $Ca^{2+}$ and $Mg^{2+}$, and addition of excess EDTA inhibited the activity of both Tce1 and DNase I (Fig. 1g and Supplementary Fig. 1d). The circular plasmid pUC19 was also assayed as a substrate (Supplementary Fig. 1e, f), further indicating that Tce1 is an endonuclease. Based on random mutant library screening, a mutant that lost toxicity to *E. coli* (Tce1$^{S8A/A16E}$) from approximately 400 candidates was identified (Fig. 1c). The purified mutant protein (Supplementary Fig. 1c) failed to cleave DNA as the Tce1 wild-type (WT) protein (Fig. 1h and Supplementary Fig. 1g), clearly demonstrating that the DNase activity of Tce1 is not due to contamination. However, Tce1 did not display detectable RNase activity in vitro (Supplementary Fig. 1h, i). Therefore, Tce1 is a $Ca^{2+}$, $Mg^{2+}$-dependent endonuclease that cleaves DNA but not RNA. Consistent with its role as an immunity protein for Tce1, addition of Tci1 to the reaction mixture effectively diminished the DNase activity of Tce1 (Supplementary Fig. 1j).

The DNase activity of YPK_0954 was further confirmed in vivo using the terminal deoxynucleotidyl transferase dUTP nick-end labeling (TUNEL) assay and DAPI staining. Although most *E. coli* cells expressing Tce1 exhibited positive TUNEL signals, indicative of DNA fragmentation, *E. coli* cells expressing Tce1$^{S8A/A16E}$ and co-expressing Tce1-Tci1, remained unlabeled, similar to the vector-only control (Fig. 1i). Furthermore, more than 80% of the *E. coli* cells expressing Tce1 lost DAPI staining after 4 h induction with IPTG. However, minimal loss of DAPI staining was observed in cells harboring empty vector, expressing Tce1$^{S8A/A16E}$, or co-expressing Tce1-Tci1 (Fig. 1j and Supplementary Fig. 2). These results establish that Tce1 is an actual DNase.

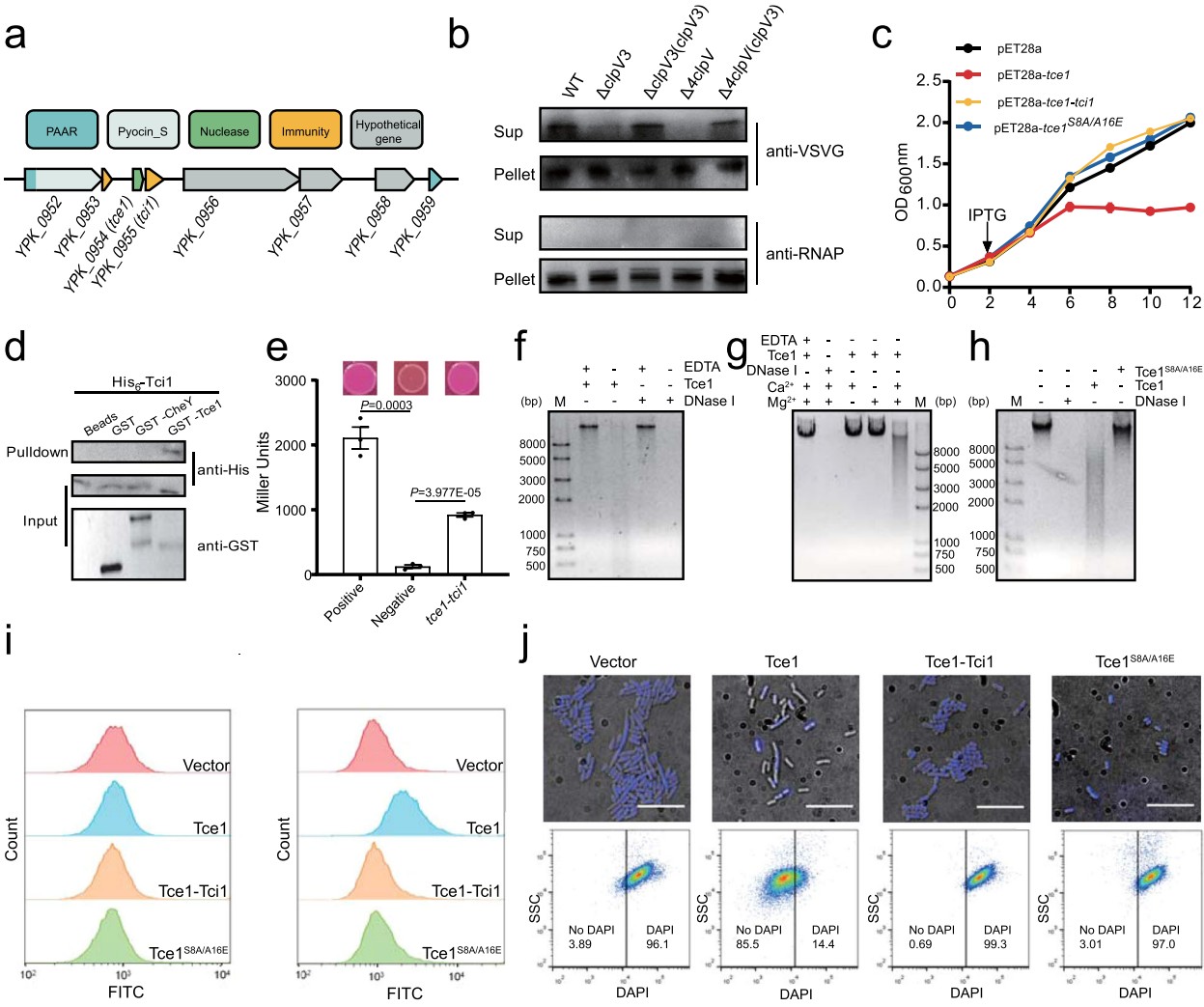

**Fig. 1 Tce1-Tci1 is a T6SS-3 nuclease effector-immunity pair. a** Schematic of genes encoding effector-immunity pairs in YPIII. **b** Tce1 is a T6SS-3 effector. A plasmid expressing Tce1-VSVG was introduced into indicated *Yptb* strains. Total cell pellet (Pellet) and secreted proteins in the culture supernatant (Sup) were isolated and probed for the presence of the fusion protein. Cytosolic RNA polymerase (RNAP) was probed as a control. **c** Growth curves of *E. coli* BL21(DE3) harboring indicated plasmids were obtained by measuring $OD_{600}$ at 2-h intervals. Data are presented as the mean ± standard deviation (SD) of three independent experiments. **d**, **e** Direct binding between Tce1 and Tci1 was detected using GST pull-down (**d**) and bacterial two-hybrid (**e**) assays. **d** His_6-Tci1 was incubated with GST-Tce1, GST, or GST-CheY, and the protein complexes captured on glutathione beads were detected using western blotting. **e** Interactions between Tce1 and Tci1 were assessed using MacConkey maltose plates (upper) and the β-galactosidase assay (lower). Error bars indicate ±SD ($n = 3$ biological replicates), two-sided, unpaired Student's *t*-test was used for these analyses, and $P < 0.05$ was considered as significant difference. **f** In vitro DNase activity assay showing integrity of λ DNA co-incubated with Tce1 or DNase I in DNase I reaction buffer with or without EDTA at 37 °C for 30 min. Reaction products were analyzed using agarose gel electrophoresis. **g** $Ca^{2+}$, $Mg^{2+}$-dependent DNase activity assay of Tce1. λ DNA was incubated with Tce1 in reaction buffer with or without $Mg^{2+}$ or $Ca^{2+}$ at 37 °C for 30 min. **h** DNase activity of the Tce1^S8A/A16E variant was tested along with Tce1 and DNase I in the presence of $Ca^{2+}$ and $Mg^{2+}$. **i** Detection of Tce1-induced genomic DNA fragmentation before (left) and 4 h after (right) IPTG induction in the TUNEL assay. DNA fragmentation was detected based on monitoring of fluorescence intensity (indicated on the *x*-axis) using flow cytometry. The counts resulting from cell sorting are indicated on the *y*-axis. **j** Detection of the loss of DNA staining (DAPI) in indicated *E. coli* cells 4 h after IPTG induction with fluorescence microscopy (upper) and flow cytometry (lower). The *x*-axis corresponds to 450H filter reading. Scale bars: 28 μm.

**Tce1 mediates contact-independent T6SS killing**. To assess the contribution of the Tce1-Tci1 effector-immunity pair to bacterial antagonism, we performed growth competition assays using labeled derivatives of *Yptb* co-cultured under conditions promoting cell contact[24]. Notably, the Tce1-Tci1 effector-immunity pair apparently underwent a duplication event based on the identification of a highly homologous gene pair *ypk_2801-ypk_2802* in the *Yptb* genome through a homology search (Supplementary Fig. 3). To simplify the analysis, we generated a deletion mutant lacking *ypk_2801-ypk_2802* as the WT for subsequent

experiments. The WT donor exhibited a 3-fold growth advantage in competition with the Δ*tce1*Δ*tci1* recipient. This growth advantage was abrogated by deletion of *tce1* from the donor, or by expression of *tci1* in the recipient. The Tce1-mediated growth advantage requires a functional T6SS-3, as deletion of *clpV3* (Δ*clpV3***, *clpV3* deleted in the Δ*ypk_2801*Δ*ypk_2802* background) completely abolished the growth advantage (Fig. 2a).

Unexpectedly, a strong competitive advantage dependent on Tce1 was also observed when cells were grown in liquid medium. As shown in Fig. 2a, donor strains possessing Tce1 and a

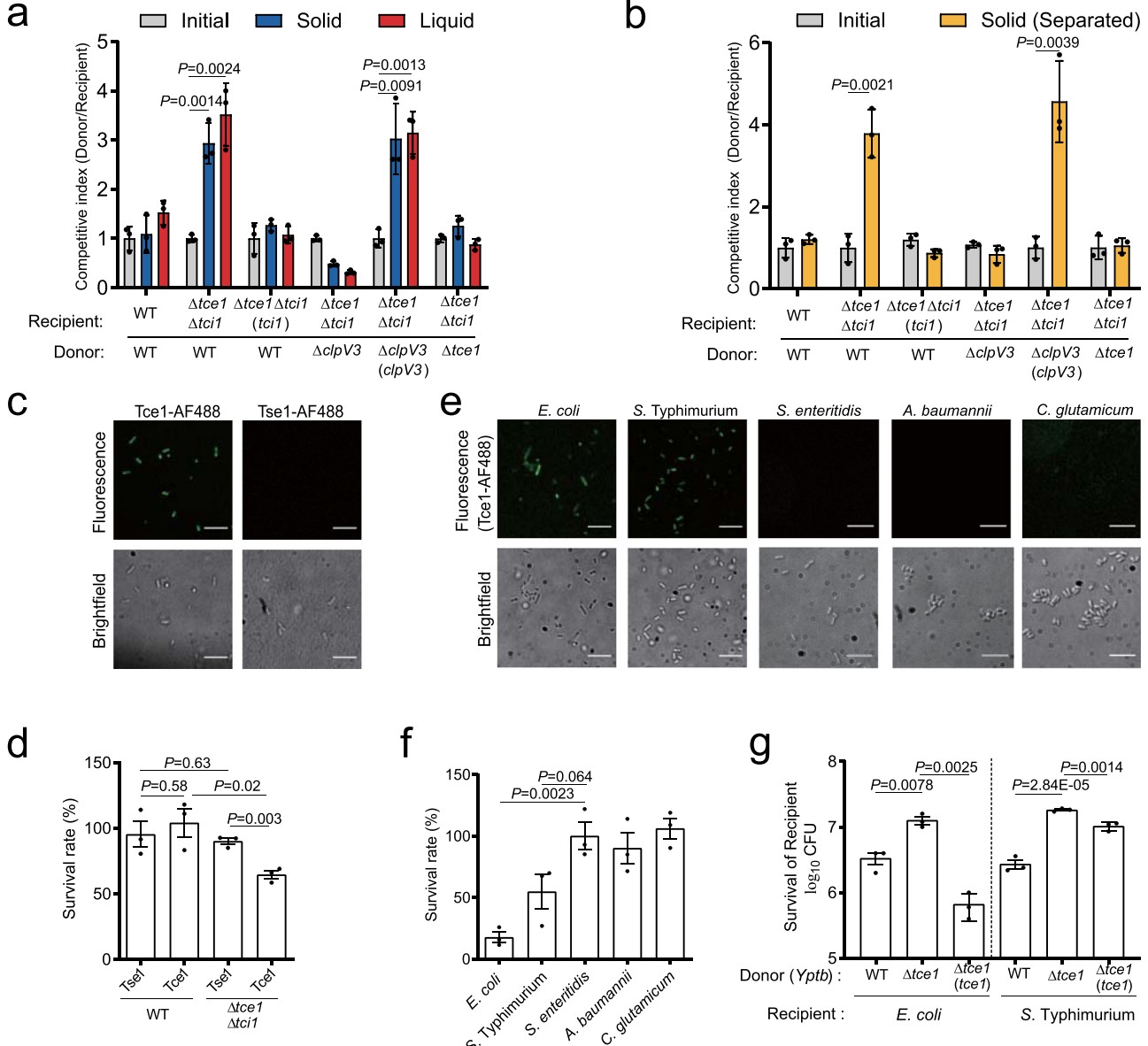

**Fig. 2 Tce1 mediates contact-independent T6SS killing. a, b** Intra-species growth competition between the indicated *Yptb* donor and recipient strains. **a** Donor and recipient strains were mixed 1:1 and then grown for 48 h on a solid support, or for 24 h in liquid medium at 26 °C. **b** Contact-independent intra-species growth competition experiments performed by separating donor and recipient cells with a cell-impermeable membrane and grown for 48 h at 26 °C on the surface of solid medium. The CFU ratio of the donor and recipient strains was measured based on plate counts. Bars represent the mean donor:recipient CFU ratio from three independent experiments (mean ± SD), with two-tailed, unpaired Student's *t*-test. Differences were considered statistically significant at $P < 0.05$. **c** Fluorescence labeling of *Yptb* YPIII with Tce1-AF488 and Tse1-AF488. (scale bars, 20 μm). Quantification of **c** was shown in Supplementary Fig. 4a. **d** Toxicity assay of purified Tce1 protein. Stationary-phase cultures of WT *Yptb* and the Δ*tce1*Δ*tci1* mutant were diluted 40-fold in M9 medium then treated with purified Tce1 or Tse1 (0.01 mg ml$^{-1}$) for 1 h, and the viability of cells was determined by counting the CFUs after treatment. **e** Fluorescence labeling of the indicated strains with Tce1-AF488 (scale bars, 20 μm). Quantification of **e** was shown in Supplementary Fig. 4b. **f** Toxicity assay of purified Tce1 protein to the indicated strains. Bacterial strains were diluted in M9 medium and treated with Tce1 (0.1 mg ml$^{-1}$) for 1 h, and the viability of cells was determined by counting the CFUs after treatment. **g** Inter-species growth competition experiments between the indicated *Yptb* donors and *E. coli* or *S.* Typhimurium recipients. Donor and recipient strains were mixed 10:1, then grown for 12 h in liquid medium at 26 °C. The survival of *E. coli* or *S.* Typhimurium cells was quantified by counting CFUs on selective plates. Data in **d**, **f**, and **g** are presented as the mean ± SD of three independent experiments. *P*-values were determined by two-tailed unpaired Student's *t*-test, and differences were acknowledged statistically significant at $P < 0.05$.

functional T6SS-3 exhibited significantly increased fitness in competition with Δ*tce1*Δ*tci1* recipients, and expression of *tci1* in the Δ*tce1*Δ*tci1* recipient rescued its competitive fitness (Fig. 2a). Similar results were obtained when the assay was repeated with a cell-impermeable membrane separating the donor and recipient cells on the surface of solid medium (Fig. 2b). Thus, T6SS-3 confers bacteria a contact-independent competitive advantage

due to secretion of an antibacterial effector into the extracellular medium, which is distinct from the canonical contact-dependent T6SS mechanism that acts as a conduit to deliver effectors across the envelope of recipient cells[25–27].

As a nuclease toxin, we assumed that Tce1 secreted into the extracellular medium must subsequently enter recipient cells to access its DNA target. To verify this prediction, we performed a

fluorescence-based assay using Alexa Fluor 488-conjugated Tce1 to probe protein importation[28]. As expected, addition of AF488-Tce1 to WT bacteria cells yielded fluorescent bacteria. By contrast, addition of AF488-conjugated Tse1, a canonical *Pseudomonas aeruginosa* T6SS toxin that requires the T6SS needle to puncture the target cell for translocation[25], did not yield fluorescent bacteria (Fig. 2c and Supplementary Fig. 4a). The contact-independent killing activity of Tce1 was also verified by examining its toxicity to target cells in liquid medium. While addition of purified Tce1 protein to the liquid medium had little effect on WT survival, it greatly reduced the survival rate of the Δtce1Δtci1 mutant, which lacks the immunity protein (Fig. 2d and Supplementary Fig. 5). Conversely, consistent with a previous report[25], the canonical T6SS toxin Tse1 exhibited no toxic effect on both bacterial strains under the same conditions (Fig. 2d). These results suggest that, unlike canonical T6SS effectors, the Tce1 effector possesses an intrinsic cell-entry mechanism. Once released into the extracellular medium, this cell-entry mechanism allows Tce1 to recognize and enter target cells independent of the T6SS needle.

Through the AF488-based protein importation assay, we found that its cell-entry mechanism also allows Tce1 to specifically enter the cytosol of *E. coli* and *Salmonella* Typhimurium, but not that of *Salmonella enteritidis*, *Acinetobacter baumannii*, or *Corynebacterium glutamicum* (Fig. 2e and Supplementary Fig. 4b). In line with these results, exogenous addition of recombinant Tce1 protein to the liquid medium substantially reduced the survival rates of *E. coli* and *S.* Typhimurium, but not of *S. enteritidis*, *C. glutamicum*, or *A. baumannii* (Fig. 2f). These findings motivated us to further investigate whether Tce1 participates in contact-independent inter-species antagonism. As expected, Tce1 significantly contributed to the fitness of *Yptb* against *E. coli* and *S.* Typhimurium during contact-independent competition assays performed in liquid medium (Fig. 2g).

Together, these results demonstrate that *Yptb* T6SS-3 follows a non-canonical contact-independent killing mechanism mediated by secretion of Tce1, a unique antibacterial effector with an intrinsic cell-entry mechanism.

**Tce1 interacts with the OM receptors BtuB and OmpF.** Bacteriocins and microcins are known to recognize specific receptors on sensitive cells to traverse the cell envelope[2,4,29]. Thus, we hypothesized that Tce1 may also interact with membrane receptors to enter target cells. To identify such putative receptors, we performed GST pull-down assays using GST-Tce1 coated beads against total cell lysates of *Yptb* WT cells. Proteins specifically retained by GST-Tce1 were detected using silver staining after sodium dodecyl sulfate–polyacrylamide gel electrophoresis (SDS–PAGE) (Fig. 3a). Mass spectrometric analysis identified several potential Tce1 partners: the 80 kDa band as BtuB, a TonB-dependent OM receptor (YPK_0782; Supplementary Fig. 6); the 40 kDa band as the OmpF porin (YPK_2649; Supplementary Fig. 7); the band around 50 kDa as TolB (YPK_2956), the periplasmic component of the TolABQR transit machinery for group A colicin importation[29]; the 38 kDa band as an OmpA domain transmembrane region-containing protein (YPK_2630); and the 90 kDa band as AcnB (aconitate hydratase 2, YPK_3487). As the involvement of BtuB, OmpF, and TolB in bacteriocin and microcin importation has been well established[29–31], we focused further investigation on BtuB, OmpF, and TolB. The specific interactions of Tce1 with BtuB, OmpF, and TolB were further supported by bacterial two-hybrid (Fig. 3b) and in vitro binding assays (Fig. 3c–e).

**Potential roles of BtuB and OmpF in Tce1 cell entry.** To test potential roles of BtuB and OmpF in Tce1 cell entry, we performed a fluorescence-based assay using Alexa Fluor 488-conjugated Tce1 to probe its import in vivo[28]. Although addition of Tce1-AF488 to WT *Yptb* and Δ*btuB* or Δ*ompF* mutants yielded fluorescent bacteria, the Δ*btuB*Δ*ompF* mutant was not labeled. Labeling of Δ*btuB*Δ*ompF* was weakly but substantially restored through complementation of *ompF*, and strongly restored through complementation of *btuB* (Fig. 4a and Supplementary Fig. 4c). In addition, we repeated the experiment using GFP labeling instead of AF488. Although exogenous GFP-Tce1 protein could not enter into the cytoplasm of target cells, it specifically labeled bacterial cells expressing *ompF* or *btuB* on the cell surface. In contrast, Δ*btuB*Δ*ompF* was not labeled with GFP-Tce1 protein (Fig. 4b and Supplementary Fig. 4d). The BtuB/OmpF-dependent entry of Tce1 into cytosol was further confirmed based on a cell fractionation experiment. As shown in Supplementary Fig. 8, Tce1 was detected in the cytosol of WT *Yptb* but not the Δ*btuB*Δ*ompF* mutant. However, complementation of *btuB* partially restored cytosolic Tce1 to the WT level.

These observations indicate that BtuB or OmpF are required for efficient uptake of Tce1 by target cells. Thus, *Yptb* with mutated BtuB or OmpF can be expected to show natural resistance to Tce1, even when the immunity protein Tci1 is not present. We tested this hypothesis by treating various mutants with different concentrations of purified Tce1 in M9 medium. As expected, deletion of *btuB* or *ompF* somewhat reduced the sensitivity of the Δtce1Δtci1 mutant to exogenously supplied Tce1 protein, while the Δtce1Δtci1Δ*btuB*Δ*ompF* quadruple mutant was not sensitive to Tce1 protein. However, the sensitivity of Δtce1Δtci1Δ*btuB*Δ*ompF* was weakly restored through complementation of *ompF*, and strongly restored through complementation of *btuB* (Fig. 4c and Supplementary Fig. 5a–c). This also explains why *E. coli* and *S.* Typhimurium, which contain highly similar BtuB and OmpF homologs, are sensitive to Tce1 treatment, but *S. enteritidis*, *C. glutamicum*, and *A. baumannii*, which do not contain such highly similar BtuB and OmpF homologs, are immune to Tce1 treatment (Supplementary Figs. 6 and 7).

Consistent with its crucial role in facilitating bacteriocin transfer across the OM[29,31], the *tolB* deletion mutant was also not sensitive to Tce1 treatment, and complementation of *tolB* restored the sensitivity to the WT level (Supplementary Fig. 9a). Moreover, while Tce1-AF488 labeled the whole cell of WT *Yptb* and the Δ*tolB*(*tolB*) complemented strain, it only weakly labeled the Δ*tolB* mutant on the cell surface (Supplementary Fig. 9b, c), further supporting that TolB plays a role in facilitating Tce1 translocation across the OM.

To further investigate whether BtuB and OmpF are involved in Tce1-mediated contact-independent T6SS killing, intra-species competition assays were performed in liquid medium. Although the WT strain strongly inhibited the growth of Δtce1Δtci1, it failed to inhibit the growth of the Δtce1Δtci1Δ*btuB*Δ*ompF* mutant. However, the reduced sensitivity in the Δtce1Δtci1Δ*btuB*Δ*ompF* mutant was substantially restored by complementation of *ompF* or *btuB* (Fig. 4d). Similar results were obtained when the assay was repeated with a cell-impermeable membrane separating the donor and recipient cells on the surface of solid medium (Supplementary Fig. 10). Similarly, the *E. coli* Δ*btuB*Δ*ompF* mutant was clearly more tolerant to WT *Yptb* attack than the WT *E. coli* in liquid medium (Fig. 4e), further supporting that both BtuB and OmpF are required for Tce1-mediated contact-independent T6SS killing.

We also examined the roles of BtuB and OmpF in Tce1-mediated contact-dependent T6SS killing on the surface of solid medium. Unexpectedly, the WT *Yptb* caused stronger inhibition of the growth of the Δtce1Δtci1Δ*btuB*Δ*ompF* mutant compared with the Δtce1Δtci1 mutant (Supplementary Fig. 11a). A potential explanation for this apparent discrepancy is that the membrane of the Δtce1Δtci1Δ*btuB*Δ*ompF* mutant is vulnerable to T6SS attack. Consistent with this possibility, we found that WT *Yptb* exhibited stronger inhibition of

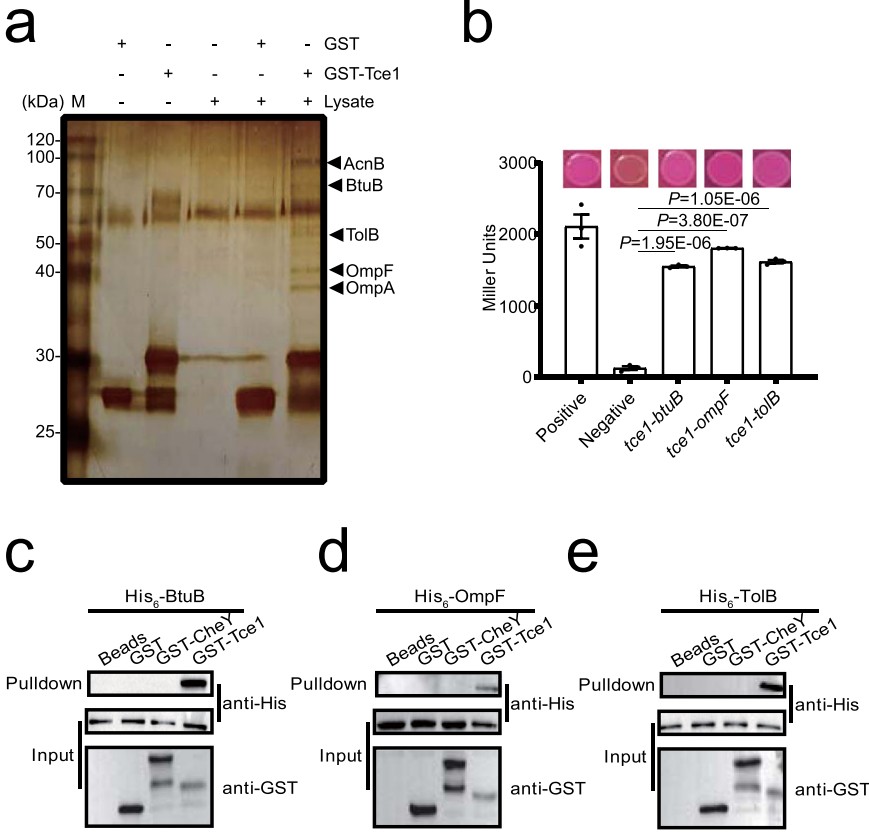

**Fig. 3 Tce1 interacts with the outer membrane receptors BtuB and OmpF. a** Identification of BtuB and OmpF as the binding partners of Tce1. Total cell lysates of *Yptb* YPIII were incubated with GST·Bind beads coated with GST-Tce1 or GST. After removing unbound proteins through extensive washing, the retained proteins were resolved through SDS–PAGE followed by silver staining. Protein bands specifically retained by GST-Tce1 were identified through mass spectrometry. **b** Interactions between Tce1 and BtuB, OmpF or TolB detected using bacterial two-hybrid assay. Interactions were assessed using MacConkey maltose plates (upper) and the β-galactosidase assay (lower). Error bars indicate ±SD ($n = 3$ biological replicates), with two-sided, unpaired Student's *t*-test. $P < 0.001$ was considered as significant differences. **c**–**e** Direct binding between Tce1 and BtuB, OmpF or TolB detected using an in vitro GST pull-down assay. His$_6$-BtuB (**c**), His$_6$-OmpF (**d**), or His$_6$-TolB (**e**) was incubated with GST-Tce1, GST, or an irrelevant recombinant protein GST-CheY, and the protein complexes captured on glutathione beads were detected using western blotting. All blots were repeated for at least three times independently with similar results.

the growth of the *E. coli* Δ*btuB*Δ*ompF* mutant compared to WT *E. coli* during contact-dependent competition (Supplementary Fig. 11b). Collectively, these results indicate that BtuB and OmpF are important for contact-independent, but not for contact-dependent, entry of Tce1.

**The Tce1-mediated T6SS killing pathway facilitates gut colonization.** Some enteric pathogens have been reported to use T6SS to kill symbionts and become established in the mammalian gut[32–34]. To investigate the role of Tce1 in facilitating *Yptb* colonization of the gastrointestinal tract, streptomycin-treated and untreated mice were orally infected with equivalent doses ($10^9$ colony-forming units, CFUs) of WT *Yptb* or the Δ*tce1* and Δ*clpV3** mutant. The degree of colonization of the cecum and small intestine of infected mice was determined at 24 and 48 h post-infection (Fig. 5a, b and Supplementary Fig. 12a, b). Without streptomycin treatment, the CFU level of Δ*tce1* and Δ*clpV3** was significantly lower compared with the WT in both organs. These results demonstrate that the Tce1-mediated T6SS killing pathway is required for colonization of the mouse gut, as it allows for outcompeting of gut commensals. This conclusion was supported by the finding that expression of the *tce1* and T6SS-3 genes was strongly induced during mouse infection compared with growth

in YLB medium (Supplementary Fig. 12c). However, pre-treatment with antibiotics greatly reduced the observed difference between the WT and Δ*tce1* in both organs, and between the WT and Δ*clpV3** in the small intestine (Fig. 5a, b), indicating that the Tce1-mediated T6SS antibacterial mechanism is not necessary to become established in the gut in the absence of an intact commensal microbial community.

*E. coli* is known to play important roles in resisting colonization of enteric pathogens in the phylum Proteobacteria[35]. Our finding that the Tce1-mediated T6SS-3 pathway targets *E. coli* in vitro prompted us to further investigate whether it can facilitate *Yptb* overcoming colonization resistance in vivo through antagonism of resident gut *E. coli*. Therefore, mice were treated with streptomycin for 24 h to reduce the number of indigenous commensal bacteria, followed by oral inoculation of mice with $5 \times 10^8$ CFUs of *E. coli* DH5α. After 24 h, mice that had been colonized with *E. coli* were challenged with $5 \times 10^8$ CFUs of WT *Yptb* or Δ*tce1*. At both 24 and 48 h after infection, the *E. coli* intestinal load of mice challenged with WT *Yptb* was significantly lower than that of mice challenged with Δ*tce1*. By contrast, WT *Yptb* exhibited significantly higher levels of colonization in mice pre-colonized with *E. coli* commensals relative to Δ*tce1* (Fig. 5c, d and Supplementary Fig. 12d, e). These results demonstrated that

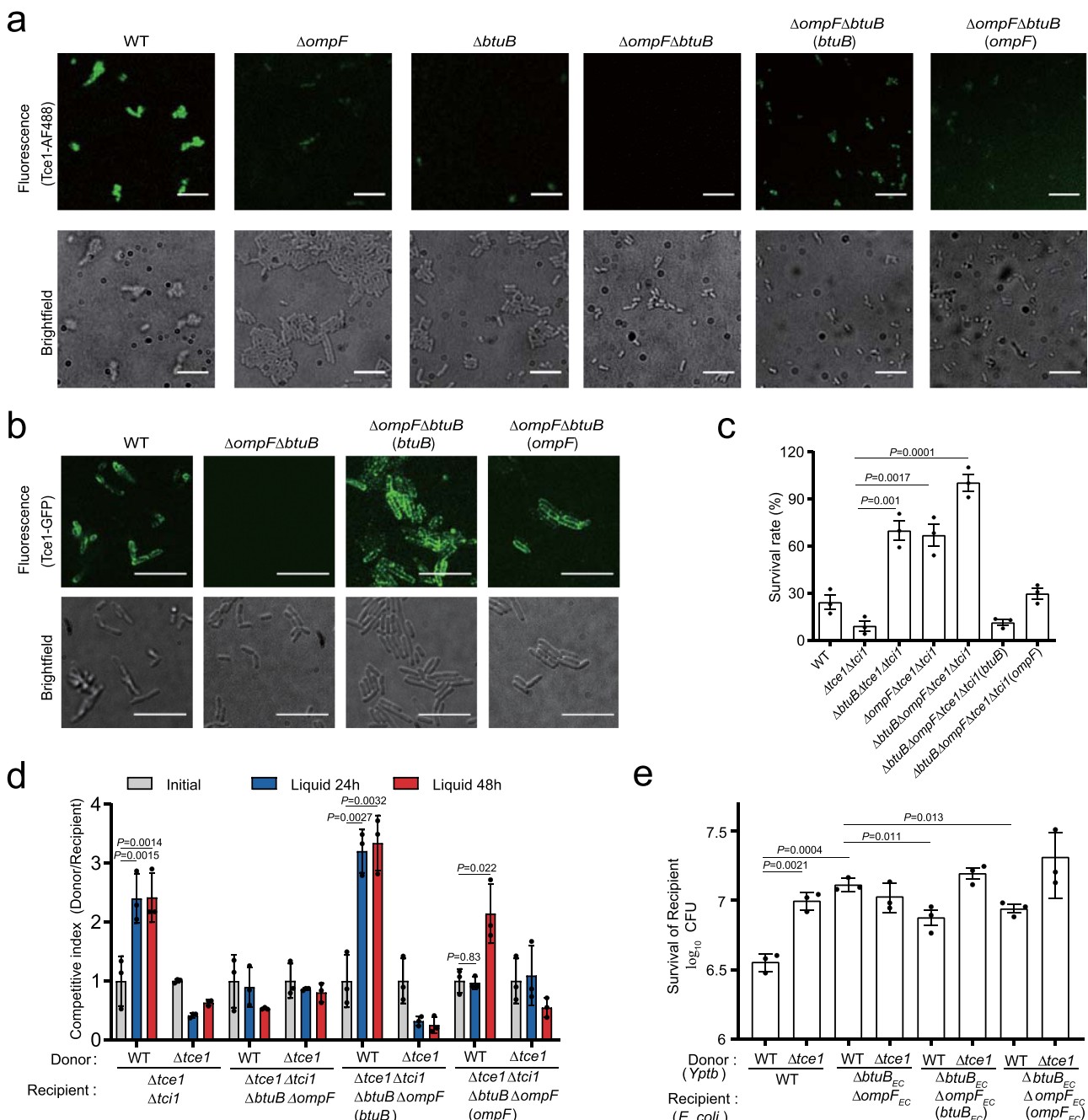

**Fig. 4 Tce1 requires BtuB and OmpF for target cell entry. a, d** Fluorescence labeling of the indicated *Yptb* strains with Tce1-AF488 (**a**) or GFP-Tce1 (**b**). Note that the Δ*btuB*Δ*ompF* mutant shows no labeling, while WT *Yptb* and complemented strains are labeled in both assays (scale bars, 20 μm). Quantification of **a** and **b** was shown in Supplementary Fig. 4c, d. **c** Toxicity assays of purified Tce1 protein to *Yptb* strains. The indicated *Yptb* strains were diluted 40-fold in M9 medium and treated with purified Tce1 (0.1 mg ml$^{-1}$) for 1 h, and the viability of cells was determined by counting the CFU after treatment. **d** Intra-species growth competition experiments between the indicated *Yptb* donor and recipient strains. Donor and recipient strains were mixed 1:1 and grown for 24 or 48 h in liquid medium at 26 °C. Bars represent the mean donor:recipient CFU ratios of three independent experiments (±SD). **e** Inter-species growth competition experiments between the indicated *Yptb* donor and *E. coli* DH5α recipient strains. Donor and recipient strains were mixed 10:1, grown for 12 h in liquid medium at 26 °C. The survival of *E. coli* cells was quantified by counting CFUs on selective plates. Data are mean ± SD from three biological replicates. *P*-values from all data were determined using two-sided, unpaired Student's *t*-test, and significant differences were considered as *P* < 0.05.

the Tce1-mediated T6SS killing pathway plays a crucial role in overcoming colonization resistance through antagonism of commensal *E. coli*.

To gain further insight into the role of Tce1 in gut colonization, we investigated whether the antibacterial activity of Tce1 is directed against enteric pathogens such as *S.* Typhimurium that

share the niche of *Yptb*. Streptomycin-treated mice were orally co-infected with $5 \times 10^8$ CFUs of WT *Yptb* or Δ*tce1* or $5 \times 10^8$ CFUs of *S.* Typhimurium. The *Yptb* and *S.* Typhimurium levels were measured in the intestine of co-infected mice at 8 h post-infection. Again, a dramatic decrease in *Salmonella* loads were observed in the cecum and small intestine of mice co-infected

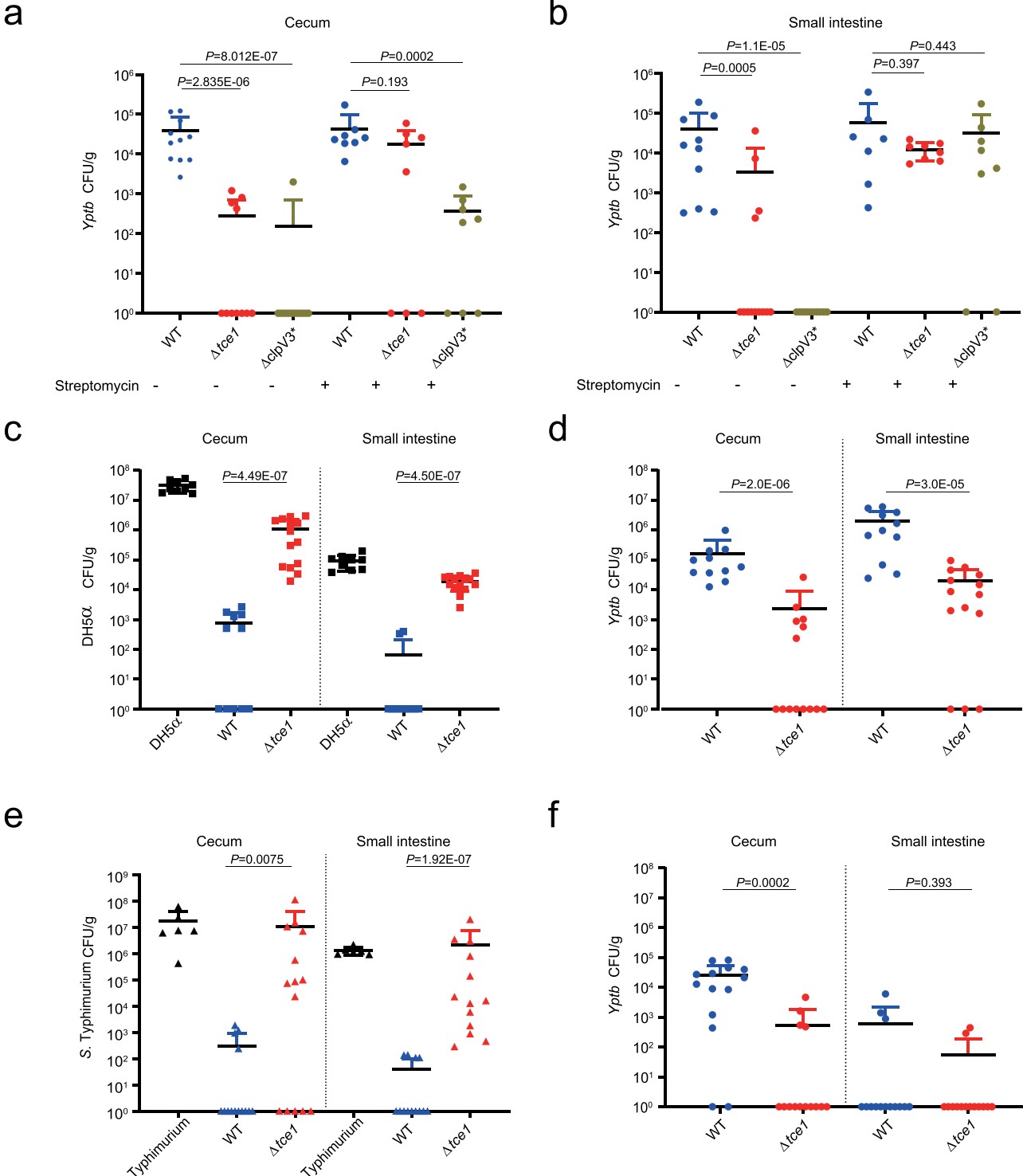

**Fig. 5 The Tce1-mediated T6SS killing pathway facilitates *Yptb* colonization of the mouse gut. a**, **b** Mice with and without streptomycin pre-treatment in **a** (Strep-: WT *n* = 11; Δ*tce1 n* = 11; Δ*clpV3*\* *n* = 13 and Strep + : WT *n* = 8; Δ*tce1 n* = 8; Δ*clpV3*\* *n* = 8) and **b** (Strep-: WT *n* = 10; Δ*tce1 n* = 13; Δ*clpV3*\* *n* = 10 and Strep + : WT *n* = 7; Δ*tce1 n* = 8; Δ*clpV3*\* *n* = 8) were orally gavaged with 10^9 CFUs of the indicated *Yptb* strains. Animals were sacrificed 24 h after bacterial challenge, and bacterial loads in cecum (**a**) and small intestine (**b**) were measured. **c**, **d** Streptomycin-treated mice (*n* = 9–14) were colonized with 5 × 10^8 CFUs of *E. coli* for 24 h and then challenged with 5 × 10^8 CFUs of WT *Yptb*, Δ*tce1* or buffer control (buffer control: *n* = 9; WT: *n* = 11; Δ*tce1*: *n* = 14). Animals were sacrificed 24 h after the challenge, and surviving *E. coli* (**c**) and *Yptb* (**d**) in the cecum and small intestine were counted. PBS mixed with *E. coli* was used as the negative control. **e**, **f** Streptomycin-treated mice were gavaged with a 1:1 mixture of *S.* Typhimurium and WT *Yptb* or Δ*tce1* mutant (buffer control: *n* = 6; WT: *n* = 14; Δ*tce1*: *n* = 14). Animals were sacrificed 8 h after the challenge, and CFU of *S.* Typhimurium (**e**) and *Yptb* (**f**) in the cecum and small intestine were counted. PBS mixed with *S.* Typhimurium was used as the negative control. Each data point represents result from one mouse; error bars represent mean ± SD of recovered CFUs. Statistical analysis of all experiments was carried out using the two-sided Mann–Whitney test. Differences were determined as significant when *P* < 0.05.

with WT *Yptb* but not with Δ*tce1*. In contrast, the WT *Yptb* levels were higher than Δ*tce1*, especially in the cecum (Fig. 5e, f), indicating that the Tce1-mediated T6SS killing pathway plays a crucial role in niche competition by targeting other enteric pathogens.

## Discussion

Here, we report a non-canonical T6SS killing pathway in *Yptb* in which a small nuclease effector, Tce1, can be translocated into the recipient cell in a contact-independent manner. This is in sharp contrast to the contact-dependent delivery by the T6SSs that directly translocate effectors across the envelope of recipient cells[25,26,36]. Unlike canonical T6SS effectors, Tce1 can enter prey cells after being secreted to the extracellular medium (Figs. 2 and 4). Interestingly, T6SS-3 can also deliver Tce1 into target cells in a contact-dependent, receptor-independent manner likely with greater efficiency (Supplementary Fig. 11). This dual mode of translocation makes T6SS-3 an advanced bacterial weapon that functions not only on solid surfaces and in biofilms, but also in liquid culture, and might protect the attacking cells from harm due to retaliatory T6SS attacks[10] from target cells.

Although we have shown that BtuB and OmpF are important for Tce1 cell entry, the underlying translocation mechanisms remain unclear. Further work is needed to test whether Tce1 cell entry is mediated by mechanisms similar to those used by some bacteriocins, which use BtuB, OmpF or other OM proteins as receptors[28–31,37]. As numerous T6SS-related toxins have been detected in growth media previously[8,24,38], it is tempting to speculate that other T6SS toxins may be able to enter target cells in a contact-independent manner.

## Methods

**Ethics statement.** All mouse experimental procedures were performed in accordance with the Regulations for the Administration of Affairs Concerning Experimental Animals approved by the State Council of People's Republic of China. The protocol was approved by the Animal Welfare and Research Ethics Committee of Northwest A&F University (protocol number: NWAFUSM2018001). Six-week-old female mice (BALB/c) were purchased from the central animal laboratory of Xi'An JiaoTong University (Xi'an, China) and kept in a temperature (24 ± 2 °C), 50 ± 10% humidity, air flow of 35 exchanges and light-controlled room (12 h light, 12 h darkness) with free access to food and water.

**Bacteria strains and growth conditions.** Bacteria strains and plasmids used in this study are listed in Supplementary Table 1. *Yptb* strains were grown in YLB (1% tryptone, 0.5% yeast extract, 0.5% NaCl) or M9 medium (Na$_2$HPO$_4$, 6 g L$^{-1}$; KH$_2$PO$_4$, 3 g L$^{-1}$; NaCl, 0.5 g L$^{-1}$; NH$_4$Cl, 1 g L$^{-1}$; MgSO$_4$, 1 mM; CaCl$_2$, 0.1 mM; glucose 0.2%) at 26 °C or 30 °C. *E. coli* and *S.* Typhimurium were cultured in LB broth at 37 °C or 26 °C. Appropriate antibiotics were included in growth medium and their corresponding concentrations are: Ampicillin (100 μg ml$^{-1}$), Nalidixic acid (20 μg ml$^{-1}$), Kanamycin (50 μg ml$^{-1}$), Tetracycline (5 μg ml$^{-1}$ for *Yptb* and 15 μg ml$^{-1}$ for *E. coli*), Gentamicin (20 μg ml$^{-1}$), Chloramphenicol (20 μg ml$^{-1}$).

**Plasmid construction.** Primers used in this study are listed in Supplementary Table 2. For obtaining expression plasmids, the genes encoding *Yptb* Tce1 (YPK_0954) was amplified by PCR. The DNA fragment was digested and cloned into similarly digested pGEX6p-1 and pET28a vectors, yielding corresponding plasmid derivatives. The expression clones of Tci1 (YPK_0955), BtuB (YPK_0782), OmpF (YPK_2649), TolB (YPK_2956) were obtained with the same method. As for the expression plasmid pET28a-*tce1-gfp*, primers *tce1*-F-*Bam*HI and *tce1*-R-*Eco*RI (TAA) were used to amplify *tce1* without termination codon TAA, and *GFP*-F-*Eco*RI and *GFP*-R-*Sal*I were used to amplify the *gfp* fragment. Digested fragments were inserted into pET28a to produce pET28a-*tce1-gfp*. The plasmid pDM4-Δ*tce1* used to construct the Δ*tce1* in-frame deletion mutant was made by overlap PCR. Briefly, the 800-bp upstream fragment and the 800-bp downstream fragment of *tce1* were amplified with primer pairs *tce1*-M1F-*Bam*HI/*tce1*-M1R and *tce1*-M2F/*tce1*-M2R-*Sal*I, respectively. The upstream and downstream PCR fragments were fused with the primer pair *tce1*-M1F-*Bam*HI/*tce1*-M2R-*Sal*I by overlap PCR. The resulting PCR products were digested with *Bam*HI and *Sal*I, and inserted into similar digested suicide plasmid pDM4 to produce pDM4-Δ*tce1*. The knock-out plasmid pDM4-Δ*tci1*, pDM4-Δ*btuB*, pDM4-Δ*ompF*, pDM4-Δ*ypk_2801-2802* were constructed with similar manners by using primers listed in Supplementary Table 2. To construct plasmids used in bacterial two-hybrid complementation

assays, the genes for testing were amplified by PCR from *Yptb* genomic DNA using appropriate primers. Amplified DNA fragments were digested with appropriate restriction enzymes, and cloned into the corresponding sites of pKT25 and pUT18C vectors, respectively. The plasmid pTargetF1-Δ*btuB$_{Ec}$* was used to construct the Δ*btuB$_{Ec}$* in-frame deletion mutant in *E. coli* DH5α. To construct pTargetF1-Δ*btuB*, upstream and downstream of gene *btuB$_{Ec}$* were amplified by PCR using primer pairs B3966-up-F/B3966-up-R and B3966-down-F/B3966-down-R respectively. The DNA fragment that code for *btuB$_{Ec}$* targeted sgRNA was made by PCR against pTargetF1 using primer pair B3966-g20-F/B3966-g20-R. Overlapping PCR was used to combine these three parts in the order of sgRNA-Up-Down and the resulting assembly was ligated to pTargetF1 pre-digested with *Spe*I/*Sal*I using Gibson Assembly. With a similar method, plasmid pTargetF1-Δ*ompF$_{Ec}$* was produced. To complement the Δ*tce1* mutant, primers *tce1*-F-*Bam*HI and *tce1*-R-*Sal*I were employed to amplify the *tce1* gene fragment from *Yptb* genomic DNA. The PCR product was digested with *Bam*HI/*Sal*I and ligated into similarly digested pKT100 to produce pKT100-*tce1*. The complementary plasmids pKT100-*tci1*, pKT100-*btuB*, pKT100-*tolB*, pKT100-*ompF*, pKT100-*btuB$_{Ec}$*, and pKT100-*ompF$_{Ec}$* were similarly constructed. Plasmid pME6032-*tce1-vsvg* was constructed for protein secretion assay. Briefly, primers *tce1*-F-*Eco*RI and *tce1*-R-VSVG-*Bgl*II were employed to amplify the *tce1* gene from *Yptb* genomic DNA. The PCR product was digested with *Eco*RI/*Bgl*II and inserted into similarly digested pME6032 to generate pME6032-*tce1-vsvg*. pME6032-*ypk_0952-vsvg* and pME6032-*tci1* were constructed in the same way. The integrity of the insert in all constructs was confirmed by DNA sequencing.

**In-frame deletion and complementation.** To construct in-frame deletion mutants, pDM4 derivatives were transformed into *Yptb* through *E. coli* S17-1 λpir-mediated conjugational mating, achieved by mixing 50 μl volumes of overnight LB cultures of *E. coli* S17-1 λpir donor with the *Yptb* parent. The mixture was spotted onto a non-selective LB plate and incubated for 16 h at 30 °C for mating. Integration of the introduced plasmid into *Yptb* chromosome by single cross-over was selected on YLB plates containing 20 μg ml$^{-1}$ chloramphenicol and 20 μg ml$^{-1}$ nalidixic acid. The chloramphenicol-resistant colonies were grown overnight in LB broth allowing for a second cross-over to occur. Selection for loss of the genome integrated *sacB*-containing plasmid was performed on YLB plates containing 20% sucrose and 20 μg ml$^{-1}$ nalidixic acid. Strains growing on this plate were tested for chloramphenicol sensitivity by parallel spotting on YLB plates containing either chloramphenicol or nalidixic acid. Chloramphenicol-sensitive and sucrose-resistant colonies were tested for deletion by PCR and confirmed by DNA sequencing[39,40]. For overexpression or complementation in relevant *Yptb* strains, the pME6032 or pKT100 derivatives were transformed into relevant *Yptb* strains by electroporation and the expression in *Yptb* was induced by adding 1 mM isopropyl β-D-1-thiogalactopyranoside (IPTG).

Clustered regularly interspaced short palindromic repeats with Cas9 (CRISPR-Cas9) system was used to construct deletion mutants in *E. coli* DH5α according to the method described previously[41], with the spectinomycin resistance gene in the pTargetF plasmid replaced by a chloramphenicol resistance gene (pTargetF1). The pTargetF1 derivatives harboring sg-upstream-downstream were electroporated in *E. coli* pre-transferred with vector pCas at 30 °C. The mutant colonies were selected on LB plates containing 50 μg ml$^{-1}$ kanamycin and 20 μg ml$^{-1}$ chloramphenicol. The plasmids pTargetF and pCas in the Δ*btuB$_{Ec}$* and Δ*ompF$_{Ec}$* mutant were successively eliminated by IPTG (1 mM) induction at 30 °C and overnight incubation at 37 °C, respectively[42]. After curing of the pTarget series and pCas plasmids, the deletion mutants were in trans complemented with pKT100 carrying their respective genes.

**Overexpression and purification of recombinant proteins.** To express and purify His$_6$ and GST-tagged recombinant proteins, pET28a and pGEX6P-1 derivatives were transformed into *E. coli* BL21(DE3) and *E. coli* XL-1 Blue, respectively. Bacteria were cultured in 5 ml LB at 37 °C to reach stationary phase and re-inoculated with a ratio of 1:100 into fresh LB, cultivated at 37 °C until OD$_{600}$ = 0.40. Then 0.2–0.5 mM IPTG was added into the growth medium, continue cultivation for another 12 h at 22 °C in a rotary shaker with a speed setting of 150 rpm. Cells were collected and disrupted by sonication and purified with the His•Bind Ni-NTA resin or GST•Bind Resin (Novagen, Madison, WI), respectively, according to manufacturer's instructions. Purified proteins were dialyzed against PBS (phosphate-buffered saline) at 4 °C overnight. To get highly purified His-tagged Tce1 and Tce1$^{S8A/A16E}$ proteins, further processes were adopted after His•Bind Ni-NTA resin purification. The eluted samples from Ni-NTA were desalted into QA buffer (20 mM Tris-HCl, 0.2 M NaCl, 10% glycerol, pH 7.5) and loaded onto HiTrap Q HP 1 ml using AKTA Pure 25 chromatography system (GE healthcare, USA). A 20 ml salt concentration gradient from 0.2 to 1 M of NaCl was performed to separate the protein samples. His-tagged Tce1 and Tce1$^{S8A/A16E}$ proteins were eluted at about 0.35 M NaCl and pooled. The recombinant proteins were over 85% purity analyzed by SDS–PAGE.

To express and purify outer membrane proteins, *E. coli* BL21(DE3) that contains the corresponding expression vector was grown in 5 ml LB at 37 °C and transferred into 500 ml LB until OD$_{600}$ reached 0.4. 0.3 mM IPTG was added and the growth condition of bacteria was shifted to 22 °C with shaking at 150 rpm. Incubated cells were collected and resuspended in binding buffer (20 mM Tris-HCl,

100 mM glycine, pH 8.3) with 6 M urea, and then it was centrifuged again to remove the residual membranes. The supernatant was purified with the His•Bind Ni-NTA resin and eluted with elution buffer. The denatured protein was mixed with refolding buffer (55 mM Tris-HCl, 0.21 mM NaCl, 0.88 mM KCl, 880 mM L-arginine, 0.5% SB-12, pH 7.0) with the ratio of 1/20 followed by 4 °C overnight incubation. The refolded protein was ultrafiltered to increase its purity and concentration and then dialyzed with buffer containing 55 mM Tris-HCl (pH 6.5), 0.21 mM NaCl, 10 mM L-arginine, and 0.5% SB-12[20]. Protein concentrations were determined by the Bradford assay with BSA (bovine serum albumin) as standard.

**GST pull-down assay**. To screen for binding partners of Tce1 with GST pull-down[20,43], 0.5 mg purified GST-Tce1 protein was incubated with 100 μl pre-washed glutathione beads for 2 h at 4 °C, then mixed with cleared cell lysates collected from 100 ml of *Yptb* culture for another 4 h. After incubation, the beads were collected and washed three times with PBS containing 300 mM NaCl, and three times with PBS containing 500 mM NaCl. Proteins binding on the beads were boiled with SDS sample buffer, resolved by SDS–PAGE and visualized by silver staining (Bio-Rad). Individual protein bands on the gel were excised, digested with trypsin and analyzed by matrix-assisted laser desorption/ionization/mass spectrometry (Voyager-DE STR, Applied Biosystems, Waltham, MA). To analyze protein interactions, purified GST fusion protein was mixed with 6×His fusion protein in PBS on a rotator for 2 h at 4 °C, and GST or an irrelevant protein CheY (BTH_II2365 in *Burkholderia thailandensis*) fused to GST were used as negative controls. After adding 40 μl of pre-washed glutathione beads slurry, binding was allowed to proceed for another 2 h at 4 °C. The beads were then washed five times with TEN buffer (100 mM Tris-Cl, 10 mM EDTA, 500 mM NaCl, pH 8.0). Retained proteins were resolved by SDS–PAGE and visualized by western blot.

**Bacterial two-hybrid assay**. To perform bacterial two-hybrid complementation assays[44,45], the pKT25 and pUT18C derivatives were co-transformed into *E. coli* BTH101 and cultured on MacConkey plate (Ampicillin 100 μg ml⁻¹, Kanamycin 50 μg ml⁻¹, IPTG 1 mM) at 30 °C. At the same time, the plasmid pKT25-zip/pUT18C-zip and pKT25/pUT18C were co-transformed into *E. coli* BTH101 to serve as positive and negative controls, respectively. Interactions were tested using MacConkey medium and a red colony color shows an interaction between proteins, while a white colony color attests the absence of interaction. Efficiencies of interactions between different proteins were quantified by measuring β-galactosidase activities in liquid cultures. In brief, overnight cultures were diluted to 1% and grown in LB broth with antibiotics at 30 °C until OD$_{600}$ reached 1.0 and β-galactosidase activities were assessed using ONPG as the substrate.

**Growth inhibition assay**. *E. coli* BL21(DE3) harboring pET28a empty vector, pET28a-*tce1*, pET28a-*tce1*$^{S8A/A16E}$, and pET28a-*tce1-tci1* were grown in LB medium. Overnight cultures were adjusted to the same OD$_{600}$ value and diluted 100-fold into LB broth containing appropriate antibiotics. After incubated at 26 °C, 180 rpm for 2 h, the expression of recombinant proteins was induced by the addition of 0.5 mM IPTG, and incubated continually under the same condition. The growth of cultures was monitored by measuring OD$_{600}$ at 2 h intervals.

**Protein toxicity assay**. Stationary-phase bacteria strains grown in YLB medium were collected, washed and diluted 40-fold into M9 medium, and treated with purified Tce1 and Tse1 toxins (0.005, 0.01, or 0.1 mg ml⁻¹) at 30 °C with shaking at 100 rpm for 60 min. After treatment, the cultures were serially diluted and plated onto YLB agar plates, and colonies were counted after 36 h growth at 30 °C. Percentage survival was calculated by dividing the number of CFU of treated cells by the number of CFU of cells without toxin treatment. All these assays were performed in triplicate at least three times.

**Protein secretion assay**. To perform protein secretion assays[46], *Yptb* strains were grown in 3 ml YLB at 30 °C and transferred into 300 ml M9 medium with 1 mM IPTG until OD$_{600}$ reached 0.60–0.65. Two milliliter of culture solution was collected and the cell pellets were resuspended in SDS–PAGE sample loading buffer. A total of 280 ml cultures was centrifuged at the speed of 5000 rpm for 20 min, and the supernatant was centrifuged for another 50 min at 9900 rpm. The final supernatant was collected and filtered with a 0.22 μm pore size filter (Millipore, MA). All the proteins were collected by filtrating through a nitrocellulose filter (BA85, Whatman, Germany) three times. The filter was dissolved in 100 μl SDS sample buffer for 15 min at 65 °C and then boiled for 10 min to recover the protein present. Protein samples of both total cell pellet and culture supernatant were resolved by SDS–PAGE and detected by western blot analysis. All samples were normalized to the OD$_{600}$ of the culture and volume used in the preparation. Secretion assays for YPK_0952 were carried out by a similar procedure.

**Western blot analysis**. Protein samples were resolved by SDS–PAGE and transferred onto polyvinylidene fluoride membranes (Millipore, MA). Then the membrane was blocked in 5% (w/v) BSA for 8 h at 4 °C, and incubated with primary antibodies at 4 °C overnight: anti-VSVG (Santa Cruz biotechnology, catalog no. sc-365019, lot number: B0916), 1:1000; anti-ICDH[47],1:6000; anti-RNAP

(Santa Cruz biotechnology, catalog no. sc-56766, lot number: F2514), 1:400; anti-His (Santa Cruz biotechnology, catalog no. sc-8036, lot number: I1018), 1:500; anti-GST (Santa Cruz biotechnology, catalog no. sc-53909, lot number: F2413), 1:500; anti-β-lactamase (Santa Cruz biotechnology, catalog no. sc-66062, lot number: 8A5.A10),1:1000. The membrane was washed five times in TBST buffer (50 mM Tris-HCl, 150 mM NaCl, 0.05% Tween 20, pH 7.4), then incubated with 1:5,000 diluted horseradish peroxidase conjugated secondary antibodies (Shanghai Genomics, catalog no. DY60203, lot number: 20614) for 4 h at 4 °C, and washed further five times with TBST buffer. Signals were detected by using the ECL plus kit (GE Healthcare, Piscataway, NJ) with a Chemiluminescence imager (Tanon 5200Multi, Beijing).

**Quantitative real-time PCR (qRT-PCR)**. Total RNA was isolated from exponentially growing strains using the RNAprep Pure Cell/Bacteria Kit (TIANGEN, Beijing, China) along with the DNase I Kit (Sigma-Aldrich, Taufkirchen, Germany). The concentration of RNA was measured by NanoDrop 2000 (Thermo Fisher Scientific, USA). The TransStart Green qPCR Super-Mix (TransGen Biotech, Beijing, China) and the Bio-Rad CFX96 Real-Time PCR Detection System (Bio-Rad, USA) was used to measure mRNA abundance in each of the samples according to manufacturer's instructions. Primers used in this study are list in Supplementary Table 2. To normalize the results, the relative abundance of 16S rRNA was used as an internal standard.

**DNase assay**. Purified Tce1 protein (0.016 μM) was incubated with λ DNA (0.35 μg, Takara, Japan, catalog no. 3010) in the reaction buffer (20 mM MES, 100 mM NaCl, 2 mM CaCl$_2$, 2 mM MgCl$_2$, pH 6.9). In all, 4 mM EDTA, 4 mM other divalent metal or other component was added in the reaction system as indicated in different experiments. The reaction of DNA hydrolysis was carried out at 37 °C for 30 min or indicated time points and the integrity of DNA was analyzed by 0.7% agarose gel electrophoresis.

**RNase assay**. Total RNA was extracted from *E. coli* TG1 and tRNA from *E. coli* MRE 600 (Roche, Germany, catalog no. 10109541001) was purchased from Sigma-Aldrich. Two micrograms of RNA was incubated with different concentrations of Tce1 in same reaction system as DNase assay at 37 °C for 30 min. The integrity of RNA was detected by 2% agarose gel.

**Fluorophore labeling of proteins**. Fluorophore labeling of proteins was performed as described[28] with minor modifications. Cysteine residues were present in the C-terminus of both Tce1 and Tse1. To prepare the proteins for subsequent labeling reactions, 5 mM DTT was used to reduce the potential disulfide bonds formed by these cysteine residues, and the reactions were conducted at room temperature for 2 h. After the reduction of disulfide bonds, DTT is removed by dialysis in 20 mM potassium phosphate (pH 7.0) and 500 mM NaCl. Labeling reactions were carried out by adding 10 mM maleimide fluorophores (Thermo Fisher Scientific, USA, catalog no. A10254) dissolved in dimethyl sulfoxide into reduced protein at a molar ratio of 5:1 (maleimide: protein), followed by 4 °C incubation in the dark overnight. The reaction was quenched by adding 2 mM DTT and dialyzed into 2 L of 20 mM potassium phosphate (pH 7.0) and 500 mM NaCl overnight at 4 °C.

**Fluorescent labeling of live bacteria**. Fluorescent labeling of live bacterial strains was performed according to described methods[28] with some modifications. Briefly, cultures at OD$_{600}$ = 0.7 were centrifuged and resuspended in M9-glucose containing 1 μM fluorophore-conjugated protein, incubated in the dark at room temperature for 30 min. The cells were washed five times to remove the free label and resuspended in 100 μl volume in M9-glucose. Ten microliters of the cell suspension was dispensed onto 1% (w/v) agarose pads on a microscope slide before sealing with a clean glass coverslip. The result was obtained by high-speed rotary disc type fluorescence confocal microscope (Andor Revolution-XD, UK).

**TUNEL (terminal deoxynucleotidyl transferase dUTP nick-end labeling) and flow cytometry analysis**. Overnight culture of *E. coli* BL21(DE3) containing the pET28a plasmid or its derivatives expressing Tce1 alone (pET28a-*tce1*) or Tce1-Tci1 together (pET28a-*tce1-tci1*) were diluted 100-fold into LB broth and incubated at 26 °C with 180 rpm shaking. After incubated at 26 °C for 2 h, the expression of toxin and immunity genes was induced by addition of 0.5 mM IPTG and continue cultivation for 4 h at 26 °C. Collected cells were washed with PBS, fixed, incubated for 5 min in PBS with 0.3% Triton X-100 and stained using One-step TUNEL cell apoptosis detection kit (Beyotime Biotechnology, China). When genomic DNA breaks, exposed 3'-OH can be labeled with green fluorescent probe FITC catalyzed by terminal deoxynucleotidyl transferase (TdT), which can be detected by flow cytometry (Beckman, CytoFLEX). Ten-thousand cells were gathered for each sample and analyzed by FlowJo_V10[48].

**DAPI staining and flow cytometry analysis**. To perform DAPI staining and flow cytometry analysis[49], overnight culture of *E. coli* BL21(DE3) containing the pET28a plasmid or its derivatives expressing Tce1 alone (pET28a-*tce1*) or Tce1-Tci1 together (pET28a-*tce1-tci1*) were diluted 100-fold into LB broth and incubated at

26 °C with 180 rpm shaking. After incubated at 26 °C for 2 h, the expression of toxin and immunity genes was induced by addition of 0.5 mM IPTG and continue cultivation for 4 h at 26 °C. Collected cells were washed with PBS, fixed, incubated for 5 min in PBS with 0.3% Triton X-100 stained using 10 μg ml$^{-1}$ DAPI for 30 min at 37 °C (Solarbio, China), then washed three times with PBS and detected by fluorescence microscope (Andor Revolution-XD, Britain) or flow cytometry (Beckman, CytoFLEX). Twenty-thousand cells were gathered for each sample and analyzed by FlowJo_V10.

**Subcellular fractionation**. To perform subcellular fractionation[50], 2 ml overnight grown Yptb culture (OD$_{600}$ 1.0) was collected, washed, and incubated in 2 ml M9 containing 0.05 mg Tce1 at 30 °C for 60 min. Tce1-treated bacterial cells were washed with PBS to remove extracellular Tce1 protein, and incubated into 285 μl sucrose buffer (20 mM PBS, pH 7.4, 20% sucrose, 2.5 mM EDTA) for 20 min at room temperature. After that, 285 μl ice-cold 0.5 mM MgCl$_2$ was added and incubated for 5 min with gentle agitation. The suspension was centrifuged at 7000 × g for 20 min at 4 °C to collect the supernatant containing periplasmic proteins (Peri). The pellet was resuspended in SDS-loading buffer and defined as cytoplasmic (Cyto). All the samples were examined by SDS–PAGE and western blotting analysis.

**Construction of mutant library by epPCR**. Error-prone PCR (epPCR) was conducted on plasmid pET28a-tce1 by using the QuickMutation™ Random Mutagenesis Kit (Beyotime Biotechnology, China) with primers tce1-F-BamHI and tce1-R-SalI according to manufacturer's instructions. The epPCR program was as follows: 94 °C for 3 min, 30 cycles of 30 s at 94 °C, 30 s at 55 °C, and 30 s at 72 °C, followed by 10 min at 72 °C final extension. The PCR products were gel-purified, digested with BamHI and SalI, and cloned into similarly digested pET28a. The ligation mixture was transformed into BL21(DE3). Transformants lost toxicity were screened in LB medium containing 0.5 mM IPTG and were further verified by cloning the mutated alleles of tce1 into new vector[51]. The mutations were identified by DNA sequencing analysis.

**Intra-species and inter-species competition in vitro**. For intra-species competition assays[24], overnight grown strains were washed and adjusted to OD$_{600}$ of 1.0 with M9 medium before mixing for competition. The initial donor-to-recipient ratio was 1:1 and the co-cultures were either spotted onto a 0.22 μm nitrocellulose membrane (Nalgene) placed on M9 agar plates at 26 °C for 48 h (for contact-dependent competition), or inoculated into 2 ml M9 medium at 26 °C with shaking for 24 h or 48 h (for contact-independent competition in liquid medium). For contact-independent competition performed on a solid surface, 5 μl of the recipient strain was spotted on 0.22 μm nitrocellulose membrane on M9 agar plates. After the bacterial solution was dried, another 0.22 μm nitrocellulose membrane was put on it and 5 μl of the donor strain was spotted on the same place of the second membrane and incubated at 26 °C for 48 h. The donor and recipient strains were labeled with pKT100 (Km$^R$) or pACYC184 (Cm$^R$), respectively, to facilitate screening on YLB plates. At indicated time points after the competition, the CFU ratio of the donor and recipient strains was measured by plate counts. Data from all competitions were analyzed using the Student's t-test, and the results shown represent the mean of one representative assay performed in triplicate.

For inter-species competition assays, overnight grown Yptb strains harboring pKT100 (Km$^R$) and E. coli (DH5α) or S. Typhimurium strains containing pBBRMCS5-GFP (Gm$^R$) or pME6032 (Tet$^R$) (gentamycin or tetracycline resistance) were washed three times with M9 medium, and adjusted to OD$_{600}$ = 1.0. Yptb strains diluted to 10-folds and target strains attenuated to 100-folds were mixed together so that the ratio of donor and recipient was 10 to 1 in M9 liquid, incubated at 26 °C with the speed of 120 rpm. After the competition, mixtures were serially diluted, counted on LB plates containing appropriate antibiotics, and the final CFU was determined.

**Murine infection and in vivo competition assays**. Female 6-week-old BALB/c mice were adapted in the lab for 3 days and orally gavaged with 10$^9$ CFUs of the indicated Yptb strains labeled by pKT100 (Km$^R$) and monitored for 24 or 48 h. When indicated, mice were orally gavaged with streptomycin (100 μl of 200 mg ml$^{-1}$ solution) 24 h prior to Yptb infection. At the end of the experiment animals were sacrificed, and the cecum and small intestine tissue were ground, plated on selective YLB antibiotic plates for CFU enumeration.

For competition assays between Yptb and E. coli in mouse gut, female 6-week-old BALB/c mice were orally gavaged with streptomycin (100 μl of 200 mg ml$^{-1}$ solution) on day 1. On day 2, 5 × 10$^8$ CFU E. coli DH5α containing GFP was gavaged, and on day 3, 5 × 10$^8$ CFU of Yptb strains was orally gavaged. After 24 h on day 4 and 48 h on day 5, mice were sacrificed and cecum and small intestine tissue were separated, serial diluted, spread on YLB (nalidixic acid, for selection of Yptb) or LB (gentamicin, for selection of E. coli) plates for CFU enumeration.

For competition assays between Yptb and S. Typhimurium, due to the difficulty of detecting S. Typhimurium in cecum tissues after 12 h, mice pre-treated with streptomycin for 2 days were orally gavaged with a mixture of equal bacterial count (5 × 10$^8$ CFU) of Yptb and S. Typhimurium containing pME6032. Eight hours later, mice were sacrificed, cecum and small intestine tissue were separated, serial

diluted, and spread on YLB (nalidixic acid, for selection of Yptb) or XLT4[52] (tetracycline, for selection of S. Typhimurium) plates for CFU enumeration.

**Statistics and reproducibility**. Statistical analyses were performed using Graph-Pad Prism Software (GraphPad Prism 7.00). All experiments were performed in at least three independent replicates. Statistical analyses of colonization assay in mice, intra-species and inter-species competition assay in mice were analyzed using two-sided Mann–Whitney test. All other experiments were analyzed using unpaired, two-tailed Student's t-test. Statistical significance is determined when $P < 0.05$. All gels, blots, and micrographs were repeated for at least three times independently with similar results.

**Reporting summary**. Further information on research design is available in the Nature Research Reporting Summary linked to this article.

## Data availability

The protein sequences are available from the Uniprot database (http://www.uniprot.org/). Other data supporting the findings of this study are included in the article and its Supplementary Information files, or from the corresponding authors upon request. Source data are provided with this paper.

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

## Acknowledgements

This work was supported by the grant of National Key R&D Program of China (2018YFA0901200 to X.S. and T.G.D.), the National Natural Science Foundation of China (31670053 and 31725003 to X.S., 31970114 and 31671292 to Y.W.), the Open Project Program of the State Key Laboratory of Pathogen and Biosecurity (SKLPBS1825 to X.S.). L.X. is supported by China Postdoctoral Science Foundation (2018M631201) and Shaanxi Postdoctoral Science Foundation (2018BSHTDZZ20). We thank Dr. Zhao-Qing Luo at Purdue University for valuable discussions and critical reading of the manuscript, Dr. Sheng Yang at Institute of Plant Physiology and Ecology, Chinese Academy of Sciences, and Dr. Quanjiang Ji at Shanghai Tech University for providing the CRISPR-Cas9 genome editing System, and Dr. Xiaodong Xu and Hao Nan at Northwest A&F University for advice and technical assistance. We also thank the Teaching and Research Core Facility at College of Life Science (Min Duan, Ningjuan Fan, Xiyan Chen and Hui Duan), and Life Science Research Core Services, NWAFU (Luqi Li) for technical support.

## Author contributions

L.S., J.P., Y.W., and X.S. conceived the study and designed experiments. Unless otherwise specified, L.S., J.P., and Y.Y. performed all of the experiments and conducted data analysis. L.S., J.P., Z.Z., and R.C. performed bacteria competition assays and data analysis. L.S., J.P., Y.Y., Z.Z., and R.C. generated strains, performed bacterial two-hybrid experimental and GST pull-down assays. Y.Y., Z.Z., R.C., S.J., C.Y., and L.S. performed protein expression and purification, and nuclease assay experiments. L.S., J.P., R.C., and C.Y. performed the confocal microscopy work and flow cytometry analysis. L.S., J.P., Z.W., and L.X. performed bioinformatics analysis. L.S., J.P., Y.Y., T.G.D., Y.W., and X.S. analyzed data and wrote the manuscript. All authors read and approved the final version of the paper.

## Competing interests

The authors declare no competing interests.
