## [Peer Review File · Nature Communications]

Editorial Note: This manuscript has been previously reviewed at another journal that is not operating a transparent peer review scheme. This document only contains reviewer comments and rebuttal letters for versions considered at *Nature Communications*. Mentions of the other journal and prior referee reports have been redacted.

Editorial Note: The table on page 7 of this Peer Review File have been redacted as indicated to remove third-party material where no permission to publish could be obtained.

Reviewer #1 (Remarks to the Author):

[Redacted] Seeing this improved version, I would like to congratulate the authors for taking on board such an extensive list of suggestions and revisions and carrying out so many additional experiments and data analyses. The revised paper is significantly improved and the vast majority of concerns have been addressed in a compelling manner.

I am happy to recommend publication in Nature Communications.

Reviewer #4 (Remarks to the Author):

I believe that the authors have done a thorough job of answering the referee comments. Saying that this version of the manuscript is better than the first is a real understatement. I feel it has now become a very strong manuscript.

Reviewer #5 (Remarks to the Author):

This manuscript proposes a contact-independent killing by a T6SS nuclease effector from *Yersinia pseudotuberculosis* in parallel to its contact-dependent killing activity. This manuscript has been much improved through the clear demonstration of nuclease activity of Tce1 which can be inhibited by Tci1. Furthermore, the inclusion of a mutant Tce1 which lacks toxicity strengthens the position that the observed killing is in fact due to Tce1. Whilst this manuscript presents a significant and compelling story there remain areas of concern detailed below.

Major Comments:

1) Figure S1c shows ITC data for the binding between Tce1 and Tci1. Whilst the signal to noise ratio of this data is very good, the fitting of the data is very poor. The ΔH (although not discussed in the manuscript) in the fit is greatly underestimated, in part due to inclusion of the first injection in the fitted data. The first injection in an ITC titration is notoriously inaccurate (in part from the mechanical action of the placing the syringe into the cell) and is typically excluded from the fit. Also, the end point of the fit does not match the end point of the titration. These two factors combined are giving an overestimate of the affinity of the interaction and also giving an inaccurate stoichiometry. I am surprised by the low error on the fitted K_d ($80 \text{ nM} \pm 2 \text{ nM}$), when by looking at the raw data by eye I would expect to see an affinity closer to the μM range. In order to obtain an accurate fit to the data I would suggest performing a control titration of Tce1 into buffer to determine any contribution of non-binding heats of dilution to the end point which can then be subtracted from the experimental data, and excluding the first titration point from the fit. In its current state the fit in Figure S1c is not appropriate for publication.

Unfortunately, these comments are equally true for the ITC data shown in Figure S8a where the fit does not represent the data. The affinity determined from the fit is likely considerably tighter (~ 10 -fold at a guess) than the binding data. The y-axis intercept (ΔH) of the fit approximates to $-8,000 \text{ kJ/mol}$ whilst the data clearly show that the true value will be more in the region of $-16,000 \text{ kJ/mol}$. In addition to this the fit completely misrepresents the end point of the titration data. Even the data in Fig S8b, although much better than Fig S1c and Fig S8a, is poorly fitted and requires improvement if to be published.

2) In the new version of the pull-down assay shown in Fig 3a an additional band is present uniquely in the GST-Tce1 + Lysate lane, running just below the 100 kDa marker. The intensity of this band exceeds that of any of the four identified proteins (BtuB, TolB, OmpF and OmpA). Have any attempts been made to identify this protein?

3) The fluorescence observed in Figure 4a is confusing. In the fields of view shown nearly all the Wt cells are fluorescently labelled, whilst for ΔompF and ΔbtuB only a small proportion of the cells are labelled, but these cells are labelled to a similar intensity seen for Wt. How can these data be explained?

4) In Figure S4c how is it possible that entry is lost with the deletion of either OmpF or BtuB, but reintroducing either of these genes back into the $\Delta\text{ompF}\Delta\text{btuB}$ strain restores entry? Assuming that the different panels in Figure 4a have been equally contrasted it appears that deletion of OmpF or BtuB prevents entry (except for a few anomalous cell) and reintroduction of either OmpF or BtuB into the $\Delta\text{ompF}\Delta\text{btuB}$ strain has little or no impact. This is further confused by Figure 4b where the $\Delta\text{ompF}\Delta\text{btuB}$ strain is not labelled by Tce1-GFP, but labelling is restored to near Wt levels (in at least some of the cells) by reintroducing either BtuB or OmpF

5) It remains unclear what the proposed role of BtuB and OmpF are in Tce1 contact independent killing. Line 36 states 'requires OmpF and BtuB', whereas line 338 states 'can use OmpF or BtuB'. If the aim is to reach TolB in the periplasm it is unclear how BtuB could facilitate this in the absence of OmpF unless TonB is also involved.

Minor Points:

1) In Figure S4 how is binding distinguished from entry to allow quantification of cell entry? This seems particularly pertinent for Fig S4d where the intensity of Tce1-GFP (which would almost certainly inhibit entry) is quantified. In these data from the fluorescence images it is clear only the membrane is being labelled and no entry is occurring.

2) On line 435 of the manuscript it is stated that the purities of Tce1 and Tce1S8A/A16E were in excess of 95%. Whilst this is believable for Tce1S8A/A16E, it seems overly optimistic for the Wt protein judging by Figure S1d where despite being loaded at a much lower concentration than Tce1S8A/A16E multiple higher molecular weight bands are visible on the gel .

3) Whilst Tce1S8A/A16E lacks cytotoxic activity there is insufficient characterisation of this mutant to claim that this is an active site mutation. These mutations could equally disrupt the protein fold or introduce a steric clash preventing DNA binding.

Response to Reviewers

Reviewer #1 (Remarks to the Author):

[Redacted] Seeing this improved version, I would like to congratulate the authors for taking on board such an extensive list of suggestions and revisions and carrying out so many additional experiments and data analyses. The revised paper is significantly improved and the vast majority of concerns have been addressed in a compelling manner.

I am happy to recommend publication in Nature Communications.

Response: We would like to thank the reviewer for the very positive comments on our study.

Reviewer #4 (Remarks to the Author):

I believe that the authors have done a thorough job of answering the referee comments. Saying that this version of the manuscript is better than the first is a real understatement. I feel it has now become a very strong manuscript.

Response: Many thanks to the reviewer for these supportive comments.

Reviewer #5 (Remarks to the Author):

This manuscript proposes a contact-independent killing by a T6SS nuclease effector from *Yersinia pseudotuberculosis* in parallel to its contact-dependent killing activity. This manuscript has been much improved through the clear demonstration of nuclease activity of Tce1 which can be inhibited by Tci1. Furthermore, the inclusion of a mutant Tce1 which lacks toxicity strengthens the position that the observed killing is in fact due to Tce1. Whilst this manuscript presents a significant and compelling story there remain areas of concern detailed below.

Response: We would like to thank the reviewer for the very positive comments on our study. We have revised our manuscript in accordance with the reviewer's comments.

Major Comments:

1) Figure S1c shows ITC data for the binding between Tce1 and Tci1. Whilst the signal to noise ratio of this data is very good, the fitting of the data is very poor. The ΔH (although not discussed in the manuscript) in the fit is greatly underestimated, in part due to inclusion of the first injection in the fitted data. The first injection in an ITC titration is notoriously inaccurate (in part from the mechanical action of the placing the syringe into the cell) and is typically excluded from the fit. Also, the end point of the fit does not match the end point of the titration. These two factors combined are giving an overestimate of the affinity of the interaction and also giving an inaccurate stoichiometry. I am surprised by the low error on the fitted K_d ($80 \text{ nM} \pm 2 \text{ nM}$), when by looking at the raw data by eye I would expect to see an affinity closer to the μM range. In order to obtain an accurate fit to the data I would suggest performing a control titration of Tce1 into buffer to determine any contribution of non-binding heats of dilution to the end point which can then be subtracted from the experimental data, and excluding the first titration point from the fit. In its current state the fit in Figure S1c is not appropriate for publication.

Unfortunately, these comments are equally true for the ITC data shown in Figure S8a where the fit does not represent the data. The affinity determined from the fit is likely considerably tighter (~ 10 -fold at a guess) than the binding data. The y-axis intercept (ΔH) of the fit approximates to $-8,000 \text{ kJ/mol}$ whilst the data clearly show that the true value will be more in the region of $-16,000 \text{ kJ/mol}$. In addition to this the fit completely misrepresents the end point of the titration data.

Even the data in Fig S8b, although much better than Fig S1c and Fig S8a, is poorly fitted and requires improvement if to be published.

Response: Thank you very much for your insightful comments and suggestions to improve our ITC results! We have performed these ITC experiments again according to your suggestions and provided improved supplementary **Figure S1c**, **S8a** and **S8b**. As you predicted, **Figure S1c** revealed a stoichiometry of

approximately ~1 (Tce1):1(Tci1) ($n = 1.25$) and a dissociation constant (K_d) of $1.40 \pm 0.80 \mu\text{M}$. And **Figure S8a** revealed a K_d of $0.24 \pm 0.20 \mu\text{M}$ and a stoichiometry of ~2 (Tce1):1(BtuB) ($n = 1.90$), **Figure S8b** revealed a K_d of $1.28 \pm 0.91 \mu\text{M}$ and a stoichiometry of ~3 (Tce1):1(OmpF) ($n = 0.296$).

2) In the new version of the pull-down assay shown in Fig 3a an additional band is present uniquely in the GST-Tce1 + Lysate lane, running just below the 100 kDa marker. The intensity of this band exceeds that of any of the four identified protein (BtuB, TolB, OmpF and OmpA). Have any attempts been made to identify this protein?

Response: Thank you for raising this important issue. We consistently identified this band (AcnB, aconitate hydratase 2, YPK_3487) in our pull-down results. Because we hypothesized that Tce1 may hijack specific membrane receptors on sensitive cells to traverse the cell envelope, we only focused our further study on BtuB, OmpF and TolB as they are well known proteins hijacked by some microcins and bacteriocins for entry into target cells. We have included AcnB in the revised **Figure 3a** and in the text as suggested, but the physiological role of Tce1-AcnB interaction needs to be investigated in the future.

3) The fluorescence observed in Figure 4a is confusing. In the fields of view shown nearly all the Wt cells are fluorescently labelled, whilst for ΔompF and ΔbtuB only a small proportion of the cells are labelled, but these cells are labelled to a similar intensity seen for Wt. How can these data be explained?

Response: Thank you very much for pointing out our mistakes in presenting the results in **Figure 4a**. Most of the labelled ΔompF and ΔbtuB cells showed weak fluorescence signals (The original micrographs are attached below for reference.). Even for those cells showing strong fluorescence signals, their signals are not as strong as that of Wt cells. But we mistakenly selected the regions that contain the brightest cells to present in **Figure 4a**. We have provided a new **Figure 4a** by selecting typical regions from the whole fields of view.

Original micrographs for creating Figure 4a. Indicated *Yptb* cells were labelled with Tce1-AF488 (scale bars, 50 μm) to measure Tce1 cell entry. The quantification of these images was shown in **Supplementary Fig. 4c**.

4) In Figure S4c how is it possible that entry is lost with the deletion of either OmpF or BtuB, but reintroducing either of these genes back into the

Δ ompF Δ btuB strain restores entry? Assuming that the different panels in Figure 4a have been equally contrasted it appears that deletion of OmpF or BtuB prevents entry (except for a few anomalous cell) and reintroduction of either OmpF or BtuB into the Δ ompF Δ btuB strain has little or no impact. This is further confused by Figure 4b where the Δ ompF Δ btuB strain is not labelled by Tce1-GFP, but labelling is restored to near Wt levels (in at least some of the cells) by reintroducing either BtuB or OmpF.

Response: Thank you very much for your insightful comments. According to our results, deletion of OmpF or BtuB greatly but not completely prevents entry, and reintroduction of either OmpF or BtuB into the Δ ompF Δ btuB strain substantially increased entry.

The explanation is that OmpF and BtuB may act cooperatively to mediate Tce1 cell entry when expressed at low levels, and could partially complement each other when expressed at high levels. Actually, BtuB and OmpF have been reported to cooperatively mediate cell entry of most E colicins and colicin A, with BtuB acts as the receptor, and OmpF acts as the translocator for cell penetration (Jakes & Cramer, 2012; Housden *et al.* 2013).

[Redacted]

(Jakes KS & Cramer WA, *Annu Rev Genet* 2012)

In addition, OmpF and the TonB-dependent transporter FpvAI have also been reported to act simultaneously as receptors and translocators for OM translocation of some other bacteriocins (el Kouhen *et al.*, 1993; Jakes &

Cramer, 2012; White *et al.* 2017). Thus, complementation of either OmpF or the TonB-dependent transporter BtuB alone in high expression levels into the $\Delta ompF \Delta btuB$ strain may allow Tce1 cell entry at a less-than-wt efficiency (observed as partially restoration of fluorescence). Although the fluorescence in the complemented strains is not fully restored, it increased substantially compared to the $\Delta btuB \Delta ompF$ double mutant especially by complementation of BtuB.

The same explanation applies to Figure S4d as well. When BtuB or OmpF was highly expressed, possibly by acting as receptors and translocators simultaneously, we observed that labelling of cell surface with Tce1-GFP is restored to near Wt levels (85% for BtuB, and 60% for OmpF, respectively. **Figure S4d**). Again, BtuB is more efficient than OmpF in recruitment of Tce1-GFP to the cell surface, consistent with our ITC results that Tce1 binds BtuB more tightly than OmpF.

We have discussed this important issue in Lines 349-381 in the discussion section.

Relevant references:

Housden NG *et al.* (2013) Intrinsically disordered protein threads through the bacterial outer-membrane porin OmpF. *Science* **340**, 1570-1574.

Jakes KS & Cramer WA. (2012) Border crossings: colicins and transporters. *Annu Rev Genet* **46**, 209-231.

el Kouhen R *et al.* (1993) Characterization of the receptor and translocator domains of colicin N. *Eur J Biochem* **214**, 635-39.

White P *et al.* (2017) Exploitation of an iron transporter for bacterial protein antibiotic import. *Proc Natl Acad Sci U S A* **114**, 12051-12056.

5) It remains unclear what the proposed role of BtuB and OmpF are in Tce1 contact independent killing. Line 36 states 'requires OmpF and BtuB', whereas line 338 states 'can use OmpF or BtuB'. If the aim is to reach TolB in the periplasm it is unclear how BtuB could facilitate this in the absence of OmpF unless TonB is also involved.

Response: We propose that normally OmpF and BtuB act cooperatively to mediate Tce1 cell entry especially when expressed in low level, with BtuB acts

as the receptor, and OmpF acts as the translocator. However, OmpF and BtuB can also act independently to mediate Tce1 cell entry when expressed in high level (though in low efficiency especially for OmpF), by acting simultaneously as receptors and translocators.

We agree that BtuB may require TonB to facilitate Tce1 entry in the absence of OmpF. Since the cell entry mechanism of Tce1 is very complex, we are regretful for not solving this issue in this manuscript, but we will work on this topic continually in the future.

Minor Points:

1) In Figure S4 how is binding distinguished from entry to allow quantification of cell entry? This seems particularly pertinent for Fig S4d where the intensity of Tce1-GFP (which would almost certainly inhibit entry) is quantified. In these data from the fluorescence images it is clear only the membrane is being labelled and no entry is occurring.

Response: Thank you very much for pointing out our mistakes in figure legend description especially for **Figure S4d**. Actually, we quantified cell entry in **Figure S4a-c**, and Tce1-GFP binding in **Figure S4d**, respectively, by quantifying the % of cells that show fluorescence above a relevant background intensity threshold as Reviewer #1 suggested. We also washed carefully to avoid nonspecific binding before quantification. In addition, it's easy to distinguish the fluorescence-binding cells (only the membrane is being labelled) from fluorescence-entered cells (the whole cell is evenly labelled) in enlarged images.

The mistake in **Figure S4d** has been corrected.

2) On line 435 of the manuscript it is stated that the purities of Tce1 and Tce1S8A/A16E were in excess of 95%. Whilst this is believable for Tce1S8A/A16E, it seems overly optimistic for the Wt protein judging by Figure S1d where despite being loaded at a much lower concentration than Tce1S8A/A16E multiple higher molecular weight bands are visible on the gel.

Response: Sorry for the imprecise description in the previous manuscript. We have calculated the purity of Tce1 by analyzing the pixel density and area using Gel Image System 1D image analysis software (Tanon). Based on the calculation, the description of protein purity was changed to: “The recombinant proteins were over 85% purity”.

3) Whilst Tce1S8A/A16E lacks cytotoxic activity there is insufficient characterisation of this mutant to claim that this is an active site mutation. These mutations could equally disrupt the protein fold or introduce a steric clash preventing DNA binding.

Response: We agree with you that whilst Tce1S8A/A16E lacks cytotoxic activity and even DNase activity, it's insufficient to claim this is an active site mutation. In this revision, we describe this mutant “lost toxicity to *E. coli*”, “failed to cleave DNA” in the manuscript.

REVIEWERS' COMMENTS

Reviewer #5 (Remarks to the Author):

This manuscript proposes a contact-independent killing by a T6SS nuclease effector from *Yersinia pseudotuberculosis* in parallel to its contact-dependent killing activity. In this version of the manuscript the authors have tried to address the issues raised in the previous round of review.

Major issues

- 1.) The ITC data in Figure S1c remains problematic. In the previous iteration of this manuscript 30 μM Tce1 was titrated into 3.5 μM Tci1, whilst now 100 μM Tce1 is titrated into 5 μM Tci1 (assuming that it is a typo in lines 573/574 where it says 5 μM Tce1 was titrated into 100 μM Tci1). It makes no sense that a relatively modest change in concentration results in binding being massively endothermic as previously shown to now being significantly exothermic in the current version of the manuscript. Have the authors performed control titrations of 100 μM Tce1 into buffer (and buffer into 5 μM Tci1) to verify that the signal change that they are monitoring is in fact a binding event?
- 2.) The ITC data in Figure S8 is again problematic. Whilst the fit matches the data much better than before I have a number of issues. Firstly, the data in S8a is significantly different from that seen in the previous version of the manuscript (the enthalpy change now appears to be in the order of -200 kJ/mol, where as previously it was in excess of -12000 kJ/mol). This brings me on to my next issue, why are the observed enthalpies so extraordinarily large? They exceed any protein-protein interaction that I am aware of by an order of magnitude (the current Figure S8b implies an enthalpy change in excess of -2000 kJ/mol).
- 3.) It requires a large leap of faith to envisage how a 67 amino acid protein can simultaneously use BtuB and OmpF to gain entry (in bacteriocins where this has been seen the proteins are an order of magnitude larger and dedicated regions are responsible for each binding interaction), whilst also encoding the ability to bind TolB on top of its ability to function as a nuclease.

Minor point. Were Tce1, BtuB and OmpF all in SB-12 detergent prior to dialysis for ITC. SB-12 having a cmc of 2-4 mM is going to have very low efficiency at equilibrating across a dialysis membrane, which would be made even worse if a low MWCO dialysis membrane was required to retain Tce1.

REVIEWERS' COMMENTS

Reviewer #5 (Remarks to the Author):

This manuscript proposes a contact-independent killing by a T6SS nuclease effector from *Yersinia pseudotuberculosis* in parallel to its contact-dependent killing activity. In this version of the manuscript the authors have tried to address the issues raised in the previous round of review.

Major issues

1.) The ITC data in Figure S1c remains problematic. In the previous iteration of this manuscript 30 μM Tce1 was titrated into 3.5 μM Tci1, whilst now 100 μM Tce1 is titrated 5 μM Tci1 (assuming that it is a typo in lines 573/574 where it says 5 μM Tce1 was titrated into 100 μM Tci1). It makes no sense that a relatively modest change in concentration results in binding being massively endothermic as previously shown to now being significantly exothermic in the current version of the manuscript. Have the authors performed control titrations of 100 μM Tce1 into buffer (and buffer into 5 μM Tci1) to verify that the signal change that they are monitoring is in fact a binding event?

Response: Thank you very much for your helpful comments. Because we have provided convincing evidences to show that Tce1 interacts with Tci1 (GST pull-down and bacterial two-hybrid results in Fig. 1d & e), and Tci1 is the cognate immunity protein for Tce1 (Fig. 1c, i, j and Fig. S1j), it's unnecessary to show the ITC data anymore. So we deleted the ITC data in Figure S1c.

2.) The ITC data in Figure S8 is again problematic. Whilst the fit matches the data much better than before I have a number of issues. Firstly, the data in S8a is significantly different from that seen in the previous version of the manuscript (the enthalpy change now appears to be in the order of -200 kJ/mol, where as previously it was in excess of -12000 kJ/mol). This brings me on to my next issue, why are the observed enthalpies so extraordinarily large? They exceed any protein-protein interaction that I am aware of by an order of magnitude (the current Figure S8b implies an enthalpy change in excess of -2000 kJ/mol).

Response: Because we have provided GST pull-down and bacterial two-hybrid results (Fig. 3b-d) to show the interactions between Tce1 and BtuB/OmpF convincingly, and the role of BtuB/OmpF in Tce1 entry into target cells were also confirmed by cell entry assay, competition assay and protein toxicity assay (Fig. 4a-e). All these results support that BtuB/OmpF are specific partners of Tce1. Thus, it's unnecessary to show the ITC data anymore. So we deleted the ITC data in Figure S8.

3.) It requires a large leap of faith to envisage how a 67 amino acid protein can simultaneously use BtuB and OmpF to gain entry (in bacteriocins where this has been seen the proteins are an order of magnitude larger and dedicated regions are responsible for each binding interaction), whilst also encoding the ability to bind TolB on top of its ability to function as a nuclease.

Response: Actually some well-defined small protein precedents of multifunctionality (microcins and non-lytic antimicrobial peptides) clearly demonstrated that it is possible for

small proteins to interact with multiple targets including membrane receptors.

Microcins are much smaller bacteriocins (less than 10 kDa) that use subtle and clever mechanisms to cross outer and inner membranes of Gram-negative bacteria by hijacking multiple outer membrane receptors (e.g. OmpF, FhuA, FepA and CirA, etc.) and inner membrane proteins (e.g. the inner membrane transporter ManY/ManZ, YejABEF and SbmA, etc.). Microcins exert potent bactericidal activities through binding essential enzymes (RNAP, ATPase, GyrB and aspartyl-tRNA synthetase) or interact with the inner membrane. For example, MccE492 recognizes FepA, Fiu, and/or Cir as receptors in the outer membrane, and the inner-membrane components ManY/ManZ of the mannose permease for internalization (Baquero et al., *Front Microbiol.* 2019; Rebuffat et al., *Biochem Soc Trans.* 2012; Mathavan, *Biochem Soc Trans.* 2012; Bieler et al., *J Bacteriol.* 2006; Novikova et al., *J Bacteriol.* 2007).

Similarly, some non-lytic antimicrobial peptides (8 to 50 amino acids) are able to traverse the bacterial membranes to act on intracellular targets, including DNA, RNA, cell wall and protein synthesis. Interestingly, some of these intracellular-acting peptides could interfere with multiple targets within the bacterial cells. Although the precise mechanism whereby non-lytic AMPs enter bacterial cells remains largely unknown, some of them (PR-39, Bac7 and apidaecin 1b) have been revealed to hijack the inner membrane protein SbmA to efficiently penetrate into Gram-negative bacteria (Cardoso et al., *Int J Mol Sci.* 2019; Le et al., *Antimicrob Agents Chemother.* 2017; Scocchi et al., *Curr Top Med Chem.* 2016).

So it's not surprising that Tce1 can interact with multiple membrane receptors to traverse the bacterial membranes, although the underlying mechanism remains unknown. We are regretful for not solving the entry mechanism of Tce1 in this manuscript (since this topic is beyond the major theme of our manuscript), but we will work on this topic continually in the future. And we have toned down all the statements regarding proposed contact-independent uptake mechanisms.

Relevant references:

- Baquero F, Lanza VF, Baquero MR, Del Campo R, Bravo-Vázquez DA. Microcins in Enterobacteriaceae: Peptide antimicrobials in the eco-active intestinal chemosphere. *Front Microbiol.* 2019; 10:2261.
- Mathavan I, Beis K. The role of bacterial membrane proteins in the internalization of microcin MccJ25 and MccB17. *Biochem Soc Trans.* 2012; 40(6):1539-43.
- Rebuffat S. Microcins in action: amazing defence strategies of Enterobacteria. *Biochem Soc Trans.* 2012; 40(6):1456-62.
- Bieler S, Silva F, Soto C, Belin D. Bactericidal activity of both secreted and nonsecreted microcin E492 requires the mannose permease. *J Bacteriol.* 2006; 188(20):7049-7061.
- Novikova M, Metlitskaya A, Datsenko K, et al. The *Escherichia coli* Yej transporter is required for the uptake of translation inhibitor microcin C. *J Bacteriol.* 2007; 189(22):8361-8365.
- Cardoso MH, Meneguetti BT, Costa BO, Buccini DF, Oshiro KGN, Preza SLE, Carvalho CME, Migliolo L, Franco OL. Non-lytic antibacterial peptides that translocate through bacterial membranes to act on intracellular targets. *Int J Mol Sci.* 2019; 20(19):4877
- Le CF, Fang CM, Sekaran SD. Intracellular targeting mechanisms by antimicrobial peptides. *Antimicrob Agents Chemother.* 2017; 61(4):e02340-16.
- Scocchi M, Mardirossian M, Runti G, Benincasa M. Non-membrane permeabilizing modes of action of antimicrobial peptides on bacteria. *Curr Top Med Chem.* 2016; 16(1):76-88.

Minor point. Were Tce1, BtuB and OmpF all in SB-12 detergent prior to dialysis for ITC. SB-12 having a cmc of 2-4 mM is going to have very low efficiency at equilibrating across a dialysis membrane, which would be made even worse if a low MWCO dialysis membrane was required to retain Tce1.

Response: Thank you for your suggestion! Actually the Tce1, OmpF and BtuB proteins have been desalted into same buffer using HiTrap Desalting 5 ml column in AKTA Pure 25 (GE Healthcare, USA). Anyway, we have deleted all ITC results.